# On the Effectiveness of Out-of-Distribution Data in Self-Supervised Long-Tail Learning

**Jianhong Bai**[1][*]**, Zuozhu Liu**[1][*]**, Hualiang Wang**[2]**, Jin Hao**[3]**, Yang Feng**[4]**,
Huanpeng Chu**[1]**, Haoji Hu**[1][†]

[1]Zhejiang University, [2]The Hong Kong University of Science and Technology,
[3]Harvard University, [4]Angelalign Technology

## Abstract

Though Self-supervised learning (SSL) has been widely studied as a promising technique for representation learning, it doesn't generalize well on long-tailed datasets due to the majority classes dominating the feature space. Recent work shows that the long-tailed learning performance could be boosted by sampling extra in-domain (ID) data for self-supervised training, however, large-scale ID data which can rebalance the minority classes are expensive to collect. In this paper, we propose an alternative but easy-to-use and effective solution, **C**ontrastive with **O**ut-of-distribution (OOD) data for **L**ong-**T**ail learning (COLT), which can effectively exploit OOD data to dynamically re-balance the feature space. We empirically identify the counter-intuitive usefulness of OOD samples in SSL long-tailed learning and principally design a novel SSL method. Concretely, we first localize the '*head*' and '*tail*' samples by assigning a tailness score to each OOD sample based on its neighborhoods in the feature space. Then, we propose an online OOD sampling strategy to dynamically re-balance the feature space. Finally, we enforce the model to be capable of distinguishing ID and OOD samples by a distribution-level supervised contrastive loss. Extensive experiments are conducted on various datasets and several state-of-the-art SSL frameworks to verify the effectiveness of the proposed method. The results show that our method significantly improves the performance of SSL on long-tailed datasets by a large margin, and even outperforms previous work which uses external ID data. Our code is available at https://github.com/JianhongBai/COLT.

## 1 Introduction

Self-supervised learning (SSL) methods (Chen et al., 2020; He et al., 2020; Grill et al., 2020) provide distinctive and transferable representations in an unsupervised manner. However, most SSL methods are performed on well-curated and balanced datasets (e.g., ImageNet), while many real-world datasets in practical applications, such as medical imaging and self-driving cars, usually follow a long-tailed distribution (Spain & Perona, 2007). Recent research (Liu et al., 2021) indicates that existing SSL methods exhibit severe performance degradation when exposed to imbalanced datasets.

To enhance the robustness of SSL methods under long-tailed data, several pioneering methods (Jiang et al., 2021b; Zhou et al., 2022) are proposed for a feasible migration of cost-sensitive learning, which is widely studied in supervised long-tail learning (Elkan, 2001; Sun et al., 2007; Cui et al., 2019b; Wang et al., 2022). The high-level intuition of these methods is to re-balance classes by adjusting loss values for different classes, i.e., forcing the model to pay more attention to tail samples. Another promising line of work explores the probability of improving the SSL methods with external data. (Jiang et al., 2021a) suggests re-balancing the class distributions by sampling external in-distribution (ID) tail instances in the wild. Nevertheless, they still require available ID samples in the sampling pool, which is hard to collect in many real-world scenarios, e.g., medical image diagnosis (Ju et al., 2021) or species classification (Miao et al., 2021).

---

[*]Equal contribution.
[†]Corresponding author.

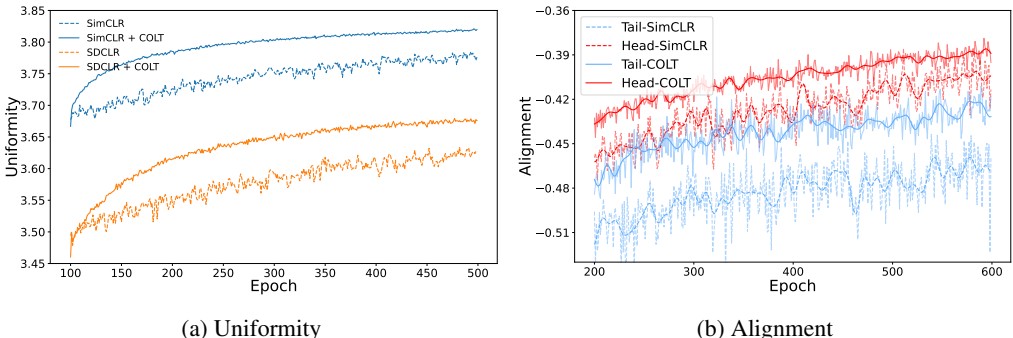

(a) Uniformity          (b) Alignment

Figure 1: (1a): Feature space uniformity of different SSL frameworks. (1b): Visualization of the alignment property of samples in minority classes and majority classes w/ or w/o COLT. The experiment is conducted with ResNet-18 on CIFAR-100-LT.

The aforementioned findings and challenges motivate us to investigate another more practical and challenging setting: *when the ID data is not available, can we leverage the out-of-distribution (OOD) data to improve the performance of SSL in long-tailed learning?* Compare to MAK Jiang et al. (2021a) that assumes external ID samples are available, while we consider a more practical scenario where we only have access to OOD data that can be easily collected (e.g., downloaded from the internet). A very recent work (Wei et al., 2022) proposes to re-balance the class priors by assigning labels to OOD images following a pre-defined distribution. However, it is performed in a supervised manner while not directly applicable to the SSL frameworks.

In this paper, we proposed a novel and principal method to exploit the unlabeled OOD data to improve SSL performance on long-tailed learning. As suggested in previous research, the standard contrastive learning would naturally put more weight on the loss of majority classes and less weight on that of minority classes, resulting in imbalanced feature spaces and poor linear separability on tail samples(Kang et al., 2020; Li et al., 2022). However, rebalancing minorities with ID samples, no matter labeled or unlabeled, is quite expensive. To alleviate these issues, we devise a framework, Contrastive Learning with OOD data for Long-Tailed learning (COLT), to dynamically augment the minorities with unlabeled OOD samples which are close to tail classes in the feature space. As illustrated in Fig. 1, our COLT can significantly improve SSL baselines in terms of the Alignment and Uniformity(Wang & Isola, 2020), two widely-used metrics to evaluate the performance of contrastive learning methods, demonstrating the effectiveness of our method.

The pipeline of our method is illustrated in Fig. 2. To augment the long-tail ID dataset, we define a tailness score to localize the head and tail samples in an unsupervised manner. Afterward, we design an online sampling strategy to dynamically re-balance the long-tail distribution by selecting OOD samples close (with a large cosine similarity in the feature space) to the head or tail classes based on a predefined budget allocation function. We follow the intuition to allocate more OOD samples to the tail classes for rebalancing. Those selected OOD samples are augmented with the ID dataset for contrastive training, where an additional distribution-level supervised contrastive loss makes the model aware of the samples from different distributions. Experimental results on four long-tail datasets demonstrate that COLT can greatly improve the performance of various SSL methods and even surpass the state-of-the-art baselines with auxiliary ID data. We also conduct comprehensive analyses to understand the effectiveness of COLT. Our contributions can be summarized as:

- We raise the question of whether we can and how to improve SSL on long-tailed datasets effectively with external unlabeled OOD data, which is better aligned with the practical scenarios but counter-intuitive to most existing work and rarely investigated before.

- We design a novel yet easy-to-use SSL method, which is composed of tailness score estimation, dynamic sampling strategies, and additional contrastive losses for long-tail learning with external OOD samples, to alleviate the imbalance issues during contrastive learning.

- We conducted extensive experiments on various datasets and SSL frameworks to verify and understand the effectiveness of the proposed method. Our method consistently outperforms baselines by a large margin with the consistent agreement between the superior performance and various feature quality evaluation metrics of contrastive learning.

## 2 RELATED WORKS

**Supervised learning with imbalanced datasets** Early attempts aim to highlight the minority samples by re-balancing strategy. These methods fall into two categories: re-sampling at the data level (Shen et al., 2016; Zou et al., 2018; Geifman & El-Yaniv, 2017), or re-weighting at the loss (gradient) level (Cao et al., 2019; Jamal et al., 2020). Due to the usage of label-related information, the above methods can not be generalized to the unsupervised field. (Kang et al., 2019) suggests that the scheme of decoupling learning representations and classifiers benefits long-tail learning. The feasibility of two-stage training promotes the exploration in unsupervised scenarios.

**Self-supervised long tail learning** (Yang & Xu, 2020) is, to our best, the first to analyze the performance of SSL methods in long-tail learning and verify the effectiveness of self-supervised pre-training theoretically and experimentally. However, (Liu et al., 2021) shows that SSL methods – although more robust than the supervised methods – are not immune to the imbalanced datasets. Follow-up studies improve the ability of SSL methods on long-tailed datasets. Motivated by the observation that deep neural networks would easily forget hard samples after pruning (Hooker et al., 2019), (Jiang et al., 2021b) proposed a self-competitor to pay more attention to the hard (tail) samples. BCL (Zhou et al., 2022) involved the memorization effect of deep neural networks (Zhang et al., 2021b) into contrastive learning, i.e., they emphasize samples from tail by assigning more powerful augmentation based on the memorization clue. We show that our method is non-conflict with existing methods and can further improve the balancedness and accuracy (Section 4.2).

**Learning with auxiliary data** Auxiliary data is widely used in the field of deep learning for different purposes, e.g., improving model robustness (Lee et al., 2020), combating label noise (Wei et al., 2021), OOD detection (Liang et al., 2018; Hendrycks et al., 2018a), domain generalization (Li et al., 2021; Liu et al., 2020; Long et al., 2015), neural network compression (Fang et al., 2021), training large models (Alayrac et al., 2022; Brown et al., 2020). In long-tail learning, MAK (Jiang et al., 2021a) suggests tackling the dataset imbalance problem by sampling in-distribution tail classes' data from an open-world sampling pool. On the contrary, we explore the probability of helping long-tail learning with OOD samples, i.e., ***none*** of the ID samples are included in the sampling pool. Open-Sampling (Wei et al., 2022) utilizes the OOD samples by assigning a label to each sample following a pre-defined label distribution. Their work is performed under supervised scenarios, and the OOD data is not filtered, which results in a massive computation overhead.

## 3 METHOD

### 3.1 PRELIMINARIES

Unsupervised visual representation learning methods aim to find an optimal embedding function $f$, which projects input image $X \in \mathbb{R}^{CHW}$ to the feature space $Z \in \mathbb{R}^d$ with $z = f(x)$, such that $z$ retains the discriminative semantic information of the input image. SimCLR (Chen et al., 2020) is one of the state-of-the-art unsupervised learning frameworks, and its training objective is defined as:

$$\mathcal{L}_{CL} = \frac{1}{N} \sum_{i=1}^{N} -\log \frac{exp(\boldsymbol{z}_i \cdot \boldsymbol{z}_i^+/\tau)}{exp(\boldsymbol{z}_i \cdot \boldsymbol{z}_i^+/\tau) + \sum_{\boldsymbol{z}_i^- \in Z^-} exp(\boldsymbol{z}_i \cdot \boldsymbol{z}_i^-/\tau)}, \tag{1}$$

where $(\boldsymbol{z}_i, \boldsymbol{z}_i^+)$ is the positive pair of instance $i$, $\boldsymbol{z}_i^-$ indicates the negative samples from the negative set $Z^-$, and $\tau$ is the temperature hyper-parameter. In practice, a batch of images is augmented twice in different augmentations, the positive pair is formulated as the two views of the same image, and the negative samples are the views of other images.

### 3.2 LOCALIZE TAIL SAMPLES IN SELF-SUPERVISED TRAINING

Due to the label-agnostic assumption in the pre-training state, the first step of the proposed method is to localize tail samples. As mentioned earlier, the majority classes dominate the feature space, and tail instances turn out to be outliers. Moreover, the minority classes have lower intra-class consistency (Li et al., 2022). Hence, a sparse neighborhood could be a reliable proxy to identify the tail samples (More analysis can be found in Section 4.4). Specifically, we use top-$k\%$ ($k = 2$

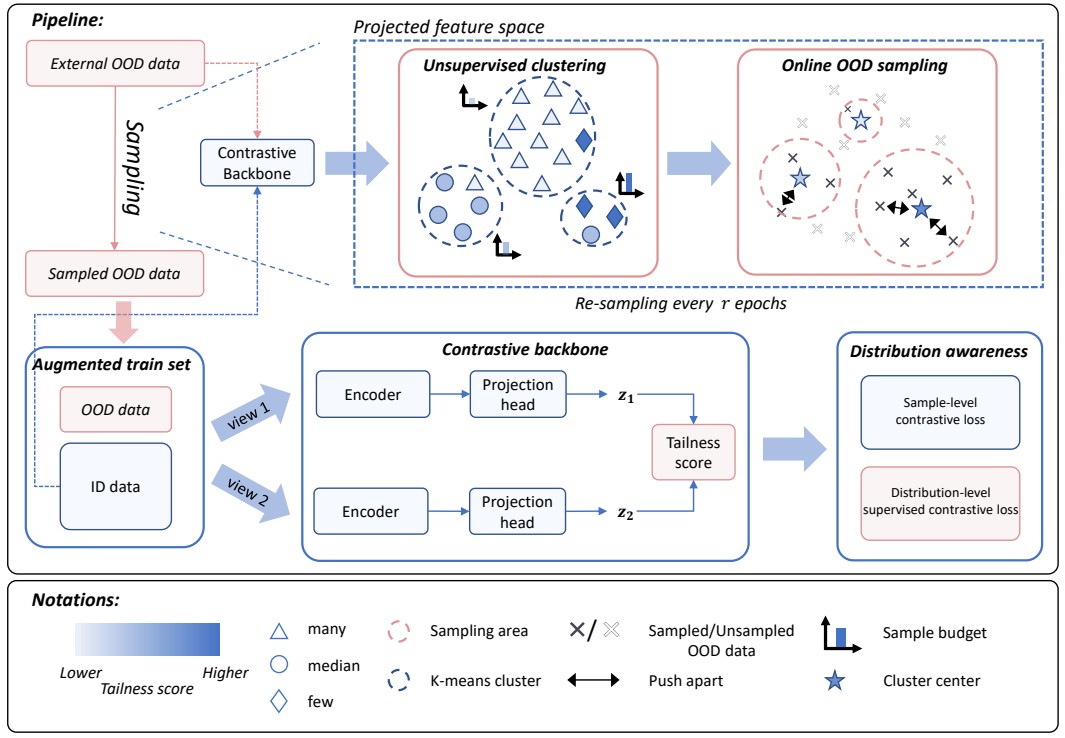

Figure 2: Overview of Contrastive with Out-of-distribution data for Long-Tail learning (COLT). COLT can be easily plugged into most SSL frameworks. Proposed components are denoted as red.

in practice) largest negative logits of each sample to depict the feature space neighborhood during training. Given a training sample $x_i$, its negative logits $p_i^-$ is the following:

$$p_i^- = \frac{exp(z_i \cdot z_i^- / \tau)}{exp(z_i \cdot z_i^+ / \tau) + \sum_{z_i^- \in Z^-} exp(z_i \cdot z_i^- / \tau)}. \tag{2}$$

Considering implementing SimCLR (Chen et al., 2020) with batch size $B$, each image has $2(B-1)$ negative samples. Then, we define $s_t^i = -\sum_{\text{top}-k\%} p_i^-$ as the tailness score for each ID instance $x_i$. During training, we perform a momentum update to the tailness score, i.e., $s_t^{i,0} = s_t^i$, $s_t^{i,n} = m s_t^{i,n-1} + (1-m) s_t^{i,n}$ where $m \in [0,1)$ is the momentum coefficient. The momentum update makes the tailness score more robust and discriminative to the tail samples. A higher value of $s_t^i$ indicates sample $x_i$ has a more sparse neighborhood in the feature space and implies that it belongs to the tail classes with a larger probability. Experiments in Fig 3e empirically demonstrated that tail samples could be effectively discovered by our proposed tailness score.

### 3.3 DYNAMICALLY RE-BALANCE THE FEATURE SPACE WITH ONLINE SAMPLING

The core of our approach is to sample OOD images from the sampling pool $S_{ood}$ and further re-balance the original long-tail ID dataset and the feature space. First, we obtain $C$ feature prototypes $z_{c_i}$ from ID training set $S_{id}$ via K-means clustering. Note that we use the features at the last projection layer since the contrastive process is performed on this layer. The cluster-wise tailness score $s_t^{c_i}$ is defined as the mean of tailness score in cluster $c_i$, i.e., $s_t^{c_i} = \sum_{z_j \in c_i} s_t^j / |c_i|$, here $|c_i|$ is the number of instances in cluster $c_i$. Then, we obtain each cluster's sampling budget $K'$ as follows:

$$K' = K \cdot softmax(\widetilde{s}_t^c / \tau_c), \quad \widetilde{s}_t^c = \frac{s_t^c - mean(s_t^c)}{std(s_t^c)}, \tag{3}$$

where $K$ refers to the total sampling budget, $K' \in \mathbb{R}^C$ is the sampling budget assigned to each cluster, $\widetilde{s}_t^c$ is the normalized cluster tailness score. Empirically, we assign more sampling budget to the tailness clusters to be consistent with the idea of re-balancing the feature space. We sample OOD images whose feature is close to (higher cosine similarity) the ID prototypes $z_{c_i}$.

To fully exploit the OOD data, we re-sample from the $S_{ood}$ every $T$ epoch. The motivation behind this is: i) the sampled OOD data can be well-separated from $S_{id}$ after a few epochs, therefore becoming less effective to re-balance the feature space; ii) over-fitting to the OOD data can be toxic to the ID performance (Wei et al., 2022). From another perspective, this online sampling strategy lets the ID training set (especially the tail samples) continuously be exposed to the more newly sampled effective negative samples, forcing the model gives more distinctive embeddings and better fitting the ID distribution. The online sampling process is summarized in Algorithm 2.

---

**Algorithm 1** The overall pipeline of COLT.

---

**Input**: ID train set $S_{id}$, OOD dataset $S_{ood}$, sample budget $K$, train epoch $T$, momentum coefficient $m$, warm-up epochs $w$, sample interval $r$, cluster number $C$, hyper-parameter $k, \tau_c$.
**Output**: pre-trained model parameter $\theta_T$.
**Initialize**: model parameter $\theta_0$, the original train set $S_{train} = S_{id}$.
    **if** $epoch = 0$ **then**
        Train model $\theta_0$ with Eq. 1 and compute $s_t^0$;
    **end if**
    **for** $epoch = 1, \cdots, T-1$ **do**
        **if** $epoch \geq w$ **then**
            **if** $(epoch - w) \% r = 0$ **then**
                Sample by performing Algorithm 2.
            **end if**
            Calculate the supervised contrastive loss with Eq. 5
            Use $S_{train}$ to train $\theta_{epoch}$ with Eq. 6;
        **else**
            Use $S_{train}$ to train $\theta_{epoch}$ with Eq. 1;
        **end if**
        Compute $s_t^i$, and update $s_t^{i,epoch}$.
    **end for**

---

**Algorithm 2** our online sampling strategy.

---

**Input**: ID train set $S_{id}$, OOD dataset $S_{ood}$, model $\theta$, sample budget $K$, cluster number $C$, similarity metric $sim(\cdot)$, hyper-parameter $\tau_c$.
**Output**: new train set $S_{train}$.
    Calculate both ID features $\mathbf{z}^{id}$ and OOD features $\mathbf{z}^{ood}$ through model $\theta$;
    Obtain $C$ ID prototypes $\mathbf{z}_{c_i}$ via K-means clustering in the projected feature space;
    Calculate cluster-wise tailness score by $s_t^{c_i} = \sum_{\mathbf{z}_j \in c_i} s_t^j / |c_i|$;
    Assign each cluster a sample budget $K'_{c_i}$ with Eq. 3;
    Initialize the sample set $S_{sample} = \emptyset$;
    **for** $i = 0, \cdots, C-1$ **do**
        Initialize subset $S_{sample}^i = \emptyset$;
        **while** $|S_{sample}| < K'_{c_i}$ **do**
            $u = \arg\max_{\mathbf{x}_j \in S_{ood}} sim(\mathbf{z}_j, \mathbf{z}_{c_i})$;
            $S_{sample}^i = S_{sample}^i \cup \{u\}$;
        **end while**
        $S_{sample} = S_{sample} \cup S_{sample}^i$;
    **end for**
    $S_{train} = S_{train} \cup S_{sample}$.

---

### 3.4 AWARENESS OF THE OUT-OF-DISTRIBUTION DATA

Section 3.2 and Section 3.3 introduce our sampling strategy toward OOD images. To involve the sampled OOD subset $S_{sample}$ in training, a feasible way is directly using the augmented training set (containing both ID and OOD samples) to train the model with Eq. 1. However, we would argue that giving equal treatment to all samples may not be the optimal choice (details in Section 4). One natural idea is to let the model be aware of that there are two kinds of samples from different domains. Hence, we define an indicator $\phi$ to provide weakly supervised (distribution only) information:

$$\phi(x_i) = \begin{cases} +1, & \mathbf{x}_i \in S_{id}; \\ -1, & \mathbf{x}_i \in S_{ood}. \end{cases} \tag{4}$$

Afterward, we add a supervised contrastive loss (Khosla et al., 2020) to both ID and OOD samples:

$$\mathcal{L}_{SCL} = \frac{1}{N} \sum_{i=1}^{N} \frac{1}{|P(i)|} \sum_{p \in P(i)} -\log \frac{exp(\mathbf{z}_i \cdot \mathbf{z}_p / \tau)}{exp(\mathbf{z}_i \cdot \mathbf{z}_p / \tau) + \sum_{n \in N(i)} exp(\mathbf{z}_i \cdot \mathbf{z}_n / \tau)}, \tag{5}$$

where $P(i) \equiv \{p : \phi(\mathbf{x}_p) = \phi(\mathbf{x}_i)\}$ is the set of indices of the same domain within the mini-batch, $|P(i)|$ is its cardinality and the negative index set $N(i) \equiv \{n : \phi(\mathbf{x}_n) \neq \phi(\mathbf{x}_i)\}$ contains index from different distribution. Fig 3c illustrates that the proposed distribution-awareness loss improves not only the overall performance but also facilitates a more balanced feature space. It's worth noting that the proposed loss only utilizes the distribution information as the supervised term, while the labels for both ID and OOD samples are unavailable during the self-supervised training stage. Finally, we scale the supervised loss with $\alpha$ and add it to the contrastive loss in Eq 1:

$$\mathcal{L}_{COLT} = \mathcal{L}_{CL} + \alpha \mathcal{L}_{SCL}. \tag{6}$$

Table 1: Test accuracy (%) and balancedness (Std↓) on CIFAR-10-LT and CIFAR-100-LT.

| Method | CIFAR-10-LT | | | | | CIFAR-100-LT | | | | |
|---|---|---|---|---|---|---|---|---|---|---|
| Metric | Many ↑ | Median ↑ | Few ↑ | Std ↓ | All ↑ | Many ↑ | Median ↑ | Few ↑ | Std ↓ | All ↑ |
| SimCLR | 82.40 | 73.91 | 70.19 | 5.11 | 75.34 | 51.50 | 45.58 | 45.96 | 2.71 | 47.65 |
| +COLT | **87.50** | **81.65** | **80.80** | **2.98** | **83.15** | **57.94** | **56.74** | **57.72** | **0.52** | **57.46** |
| SDCLR | 86.69 | 82.15 | 76.23 | 4.28 | 81.74 | 58.54 | 55.70 | 52.10 | 2.64 | 55.48 |
| +COLT | **90.87** | **84.28** | **81.45** | **3.95** | **85.41** | **63.28** | **60.85** | **59.42** | **1.59** | **61.18** |
| BCL-I | 86.97 | 82.40 | 76.45 | 4.31 | 81.99 | 58.92 | 54.63 | 53.58 | 2.31 | 55.70 |
| +COLT | **89.03** | **85.10** | **80.36** | **3.55** | **84.86** | **61.12** | **57.03** | **55.82** | **2.27** | **57.98** |

Table 2: Test accuracy (%) and balancedness (Std↓) on ImageNet-100-LT and Places-LT.

| Method | ImageNet-100-LT | | | | | Places-LT | | | | |
|---|---|---|---|---|---|---|---|---|---|---|
| Metric | Many ↑ | Median ↑ | Few ↑ | Std ↓ | All ↑ | Many ↑ | Median ↑ | Few ↑ | Std ↓ | All ↑ |
| SimCLR | 70.96 | 65.33 | 61.89 | 3.74 | 67.08 | 40.02 | 46.61 | 49.38 | 3.93 | 44.78 |
| +COLT | **75.13** | **71.38** | **66.62** | **3.48** | **72.22** | **41.55** | **48.40** | **50.54** | **3.83** | **46.36** |
| SDCLR | 71.13 | 66.04 | 62.31 | 3.61 | 67.54 | 40.13 | 46.61 | 48.90 | **3.71** | 44.73 |
| +COLT | **75.13** | **70.25** | **67.69** | **3.08** | **71.82** | **41.72** | **48.42** | **50.78** | 3.84 | **46.47** |

## 4 EXPERIMENTS

In this section, we first introduce the datasets and experimental settings (Section 4.1) and evaluate the proposed COLT in three aspects: accuracy and balancedness(Section 4.2), versatility and complexity (Section 4.3). Then, we verify whether our method can **1),** localize tail samples, **2),** re-balance the feature space. Finally, we provide a comprehensive analysis of COLT (Section 4.4).

### 4.1 DATASETS AND SETTINGS

We conduct experiments on four popular datasets. **CIFAR-10-LT/CIFAR-100-LT** are long-tail subsets sampled from the original CIFAR10/CIFAR100 (Cui et al., 2019a). We set the imbalance ratio to 100 in default. Following (Wei et al., 2022), we use 300K Random Images (Hendrycks et al., 2018b) as the OOD dataset. **ImageNet-100-LT** is proposed by (Jiang et al., 2021b) with 12K images sampled from ImageNet-100 (Tian et al., 2020) with Pareto distribution. We use ImageNet-R (Hendrycks et al., 2021) as the OOD dataset. **Places-LT** (Liu et al., 2019) contains about 62.5K images sampled from the large-scale scene-centric Places dataset (Zhou et al., 2017) with Pareto distribution. Places-Extra69 (Zhou et al., 2017) is utilized as the OOD dataset.

**Evaluation protocols** To verify the balancedness and separability of the feature space, we report performance under two widely-used evaluation protocols in SSL: *linear-probing* and *few-shot*. For both protocols, we first perform self-supervised training on the encoder model to get the optimized visual representation. Then, we fine-tune a linear classifier on top of the fixed encoder. The only difference between linear-probing and few-shot learning is that we use the full dataset for linear probing and 1% samples of the full dataset for few-shot learning during fine-tuning.

**Measurement metrics** As a common practice in long tail learning, we divide each dataset into three disjoint groups in terms of the instance number of each class: {*Many, Median, Few*}. By calculating the standard deviation of the accuracy of the three groups, we can quantitatively analyze the balancedness of a feature space (Jiang et al., 2021b). The linear separability of the feature space is evaluated by the overall accuracy.

**Training settings** We evaluate our method with SimCLR (Chen et al., 2020) framework in default. We also conduct experiments on several state-of-the-art methods in self-supervised long tail learning (Jiang et al., 2021b; Zhou et al., 2022). We adopt Resnet-18 (He et al., 2016) for small datasets (CIFAR-10-LT/CIFAR-100-LT), and Resnet-50 for large datasets (ImageNet-100-LT/Places-LT), respectively. More details can be found in Appendix.

Table 3: Compare the proposed COLT with random sample and MAK under the same sampling pool and sampling budget. The best performance under each setting is marked as **bold**.

| ID dataset | Sampling pool | Budget | Method | Protocol | Many ↑ | Median ↑ | Few ↑ | Std ↓ | All ↑ |
|---|---|---|---|---|---|---|---|---|---|
| Image Net-100 | Image Net-R | 5K | random | few-shot | 51.76 | 41.45 | 38.12 | 5.81 | 45.04 |
| | | | | linear-probing | 72.22 | 66.49 | 63.45 | 3.64 | 68.33 |
| | | | MAK | few-shot | 52.49 | 43.54 | 39.00 | 5.65 | 46.48 |
| | | | | linear-probing | 73.24 | 67.62 | 64.14 | 3.75 | 69.36 |
| | | | COLT | few-shot | **54.10** | **46.01** | **42.32** | **4.92** | **48.69** |
| | | | | linear-probing | **74.52** | **69.67** | **67.53** | **2.92** | **71.28** |
| | | 10K | random | few-shot | 52.82 | 42.88 | 40.27 | 5.41 | 46.42 |
| | | | | linear-probing | 74.33 | 68.52 | 62.65 | 4.77 | 70.02 |
| | | | MAK | few-shot | 54.33 | 45.01 | 40.12 | 5.90 | 48.01 |
| | | | | linear-probing | **75.57** | 68.20 | 66.29 | 4.07 | 70.83 |
| | | | COLT | few-shot | **54.26** | **46.54** | **43.38** | **4.57** | **49.14** |
| | | | | linear-probing | 75.13 | **71.38** | 66.62 | 3.48 | **72.22** |
| Places 365 | Places 69 | 10K | random | few-shot | 30.61 | 33.94 | 37.55 | 2.83 | 33.45 |
| | | | | linear-probing | 40.21 | 47.59 | 50.51 | 4.33 | 45.51 |
| | | | MAK | few-shot | 30.41 | 34.47 | **37.59** | 2.94 | 33.62 |
| | | | | linear-probing | 40.83 | 47.78 | 50.72 | 4.15 | 45.86 |
| | | | COLT | few-shot | **31.04** | **34.65** | 37.49 | **2.64** | **33.91** |
| | | | | linear-probing | **41.55** | **48.40** | **50.54** | 3.83 | **46.36** |

Table 4: Compare the test accuracy (%) on ImageNet-100-LT of the proposed COLT with MAK which use ID data. The best performance is marked as **bold**.

| Method | Extra type | Sample set | Many ↑ | Median ↑ | Few ↑ | Std ↓ | All ↑ |
|---|---|---|---|---|---|---|---|
| MAK | ID | IN-900 | **75.7±0.5** | 70.4±0.6 | 66.9±0.6 | 3.0±0.4 | 72.0±0.5 |
| | ID & OOD | IPM | 74.7±0.2 | 69.2±0.7 | 66.6±0.7 | 3.3±0.3 | 71.1±0.5 |
| | OOD | ImageNet-R | 75.6±0.4 | 68.2±0.8 | 66.3±0.8 | 4.1±0.6 | 70.8±0.5 |
| COLT | OOD | ImageNet-R | 75.3±0.3 | **70.9±0.8** | **69.5±0.3** | **2.4±0.7** | **72.4±0.3** |

## 4.2 COLT'S ACCURACY, BALANCEDNESS AND VERSATILITY

The main results of the proposed approach in various datasets and settings are presented in Table 1 and Table 2. We sample $K = 10,000$ OOD images on every $r = 25$ epoch for CIFAR-10-LT/CIFAR-100-LT, Places-LT, and $r = 50$ for ImageNet-100-LT. COLT significantly outperforms the baseline (vanilla SimCLR) by a large margin (about 10% for long-tail CIFAR, 5% for ImageNet-100-LT, 1.6% for Places-LT). Besides, the performance gain of the minority classes (*Median & Few*) is more notable (e.g., about 12% for long-tailed CIFAR-100). Meanwhile, COLT yields a balanced feature space. Following previous works (Jiang et al., 2021b) (Zhou et al., 2022), we measure the balancedness of a feature space through the accuracy's standard deviation from *Many, Median* and *Few*. COLT significantly narrows the performance gap between the three groups (much lower Std), which indicates we learn a more balanced feature space.

To evaluate the versatility of COLT, we carry out experiments on top of several improved SSL frameworks for long-tail learning, i.e., SDCLR (Jiang et al., 2021b) and BCL (Zhou et al., 2022). Table 1 and Table 2 also summarized COLT performance on these two methods. We can observe that incorporating our method into existing state-of-the-art methods can consistently improve their performance, which indicates that our method is robust to the underlying SSL frameworks.

## 4.3 COLT VS BASELINES WITH AUXILIARY DATA

We also compare COLT with methods that make use of external data. MAK (Jiang et al., 2021a) is the state-of-the-art method that proposes a sampling strategy to re-balance the training set by sample

Table 5: Comparison of semi-supervised and self-supervised methods when leveraging OOD data.

| Method | ID-Supervised | FixMatch | FlexMatch | ABC | DARP | SimCLR | SimCLR +COLT(10K) |
|---|---|---|---|---|---|---|---|
| Accuracy | 44.13 | 47.38 | 50.40 | 51.22 | 50.94 | 47.65 | 57.46 |

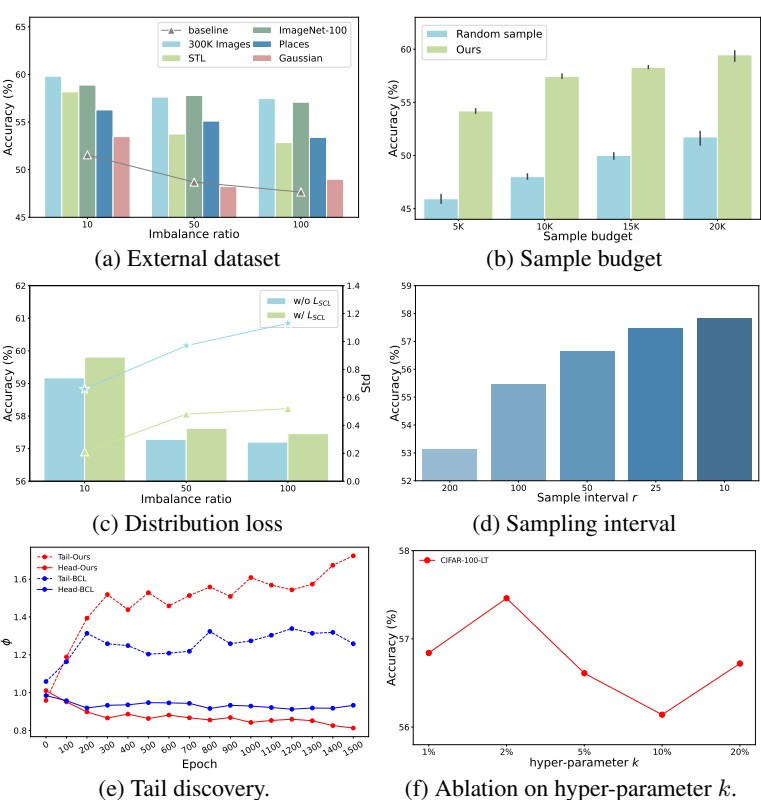

(a) External dataset  (b) Sample budget

(c) Distribution loss  (d) Sampling interval

(e) Tail discovery.  (f) Ablation on hyper-parameter $k$.

Figure 3: Analytical experiments of COLT on CIFAR-100-LT. (3a): accuracy when changing the external OOD dataset. (3b): accuracy when sampling different numbers of OOD samples on 300K Random Images. (3c): Top-1 accuracy and standard derivation (Std) of COLT with or without the proposed distribution loss. (3d): accuracy with various sampling intervals $r$. (3e): A higher $\phi_{tail}$ and a lower $\phi_{head}$ implies mining tail samples more precisely. (3f): accuracy with various $k$.

in-distribution tail class instances from an open-world sampling pool. We compare the proposed COLT with MAK in Table 3, noting that "random" refers to random sampling from the external dataset according to the budget. We observe both higher accuracy and balancedness under different sample budgets on ImageNet-100 and Places. It indicates COLT leverages OOD data in a more efficient way. Furthermore, we ask the question that *whether OOD samples can replace ID samples to help long-tail learning.* We obtain a positive answer from empirical results in Table 4. We compare the result of COLT and MAK on auxiliary data which involve ID samples. COLT achieves better performance on most of the metrics, even compared with sampling in an entirely ID dataset.

On the other, COLT has less computational overhead. MAK applies a three-stage pipeline: pre-train the model with ID samples, sample from the sampling pool, and re-train a model from the beginning. In contrast, COLT samples during the training process, resulted in a single-stage pipeline. The online sampling strategy not only fully utilized the external datasets but also reduced the computation overhead significantly[1]. Open-Sampling (Wei et al., 2022) also uses OOD data to help long-tail learning. Different from ours, they use an large data budget (300K for CIFAR), while COLT improves the baselines with a much smaller budget (10K for CIFAR).

### 4.4 ANALYSIS AND ABLATION STUDY

**The choice of OOD dataset** We conduct experiments on CIFAR-100-LT and replace the OOD dataset while maintaining other settings unchanged. As shown in Fig 3a, our method improves the ID accuracy when using 300K Random Images (Hendrycks et al., 2018b), STL (Coates et al., 2011), ImageNet-100, Places with 9.81%, 5.19%, 9.43%, 5.74% respectively. Besides, sampling on Gaussian Noise provides limited help (less than 1%) or degradation to ID accuracy.

**The effectiveness of distribution-awareness loss** COLT introduces a supervised contrastive loss to explicitly separate samples from different distributions. We conduct an ablation study on the proposed loss in Fig 3c, and the results show that the proposed loss not only improves the overall accuracy but also significantly alleviates the imbalance (i.e., much lower std).

**The effect of sampling budget** In Fig 3b, we compare the performance gains of COLT under different budgets. COLT consistently outperforms the random sampling strategy, i.e., leveraging OOD samples more effectively. Moreover, though a larger budget will give better performance, the performance gain almost plateaus with a budget of 10-15k in COLT, indicating better data efficiency.

**Comparison with semi-supervised methods** Performing semi-supervised learning is also a natural choice for utilizing external unlabeled data. We implement FixMatch (Sohn et al., 2020), FlexMatch (Zhang et al., 2021a), ABC (Lee et al., 2021), and DARP (Kim et al., 2020) on long-tailed CIFAR-100, the first two are general semi-supervised methods, and the last two are elaborately designed for long-tail learning. In Table 5, we compare the results in such semi-supervised learning scenarios (labeled: CIFAR-100-LT, unlabeled: 300K Random Images) to supervised, self-supervised, and COLT. It can be observed that 1), external unlabeled OOD data can also be helpful when performing semi-supervised learning 2), the performance gains of COLT (about 10%) are more significant than incorporating OOD data via semi-supervised training. This could be attributed to most semi-supervised methods considering unlabeled data is also ID. It may need some special design for unlabeled OOD data, e.g., resist some "toxic" samples or redesign the pseudo labels for OOD data.

**Changing of hyper-parameters** We also show the effect of hyper-parameters involved in COLT. Fig 3d shows the empirical results of changing resample interval $r$. We can observe that reasonably small intervals lead to higher accuracy. Fig 3f shows the classification accuracy when changing $k$. The limited fluctuations in performance prove that COLT is robust to hyper-parameters.

**The ability of tail sample mining** Recall in Section 3.2, we localize tail samples by assigning a predefined "tailness score" to each sample. In order to verify the effectiveness of *tailness score*, we select the top 10% samples with the highest tailness score as a subset and calculate the ratio of the percentage of {*Major* / *Minor*} samples in this subset to the percentage of whole dataset: $\phi = \frac{\mathcal{T} \cap S_{id}^{sub}}{\mathcal{T} \cap S_{id}}$ where $\mathcal{T}$ denotes the target group, $S_{id}$ is the whole in-distribution dataset, $S_{id}^{sub}$ is the subset of samples which have top $\gamma\%$ highest tailness score, $\gamma$ is set to 10. $\phi$ reflects the ability to identify tail samples: when the target group is *Minor/Major*, higher/lower $\phi$ indicates a method localizes tail samples well. As illustrated in Fig 3e, COLT discovers more samples from the tail than BCL.

## 5 CONCLUSION AND LIMITATIONS

In this paper, we propose a novel SSL pipeline COLT, which is, to our best, the first attempt to extend additional training samples from OOD datasets for improved SSL long-tailed learning. COLT includes three steps, unsupervised localizing head/tail samples, re-balancing the feature space by online sampling, and SSL with additional distribution-level supervised contrastive loss. Extensive experiments show that our method significantly and consistently improves the performance of SSL on various long-tailed datasets. There are nevertheless some limitations. First, more theoretical analyses are needed to better understand the effectiveness of OOD samples. Besides, for a given long-tail ID dataset, how to specify the best OOD dataset that gives the largest improvements is also worth exploring. We hope our work can promote the exploration of OOD data in long-tail scenarios.

---

[1]Although we sample for multiple times (every $r$ epochs) in COLT, the time of performing sampling once is less than training for one epoch; therefore can be ignored compared to 1.7x training epochs brought by MAK.

ACKNOWLEDGEMENT

This work is supported by the National Natural Science Foundation of China (Grant No. U21B2004, 62106222), the Zhejiang Provincial key RD Program of China (Grant No. 2021C01119), the Natural Science Foundation of Zhejiang Province, China(Grant No. LZ23F020008), the Core Technology Research Project of Foshan, Guangdong Province, China (Grant No. 1920001000498) and the Zhejiang University-Angelalign Inc. R&D Center for Intelligent Healthcare. Jianhong Bai would also like to thank Huan Wang from Northeastern University (Boston, USA), Denny Wu from University of Toronto, and Hangxiang Fang from Zhejiang University for their guidance and help.

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

## APPENDIX A  DATASETS AND TRAINING DETAILS

**CIFAR-10-LT/CIFAR-100-LT** are first introduced by (Cui et al., 2019a), which are long-tail subsets sampled from the original CIFAR10/CIFAR100. The imbalance ratio is defined as the instance number of the largest class divided by the smallest class. To better reflect the performance difference, we set the imbalance ratio to 100 in default. Following (Wei et al., 2022), we use 300K Random Images (Hendrycks et al., 2018b) [2] as the OOD dataset. In addition, we also conduct experiments with OOD datasets as STL-10 (Coates et al., 2011), which contains 5,000 labeled images and 100,000 unlabeled images in 10 classes with a resolution of 96x96. We use all the unlabeled images as external OOD data.

**ImageNet-100-LT** is proposed by (Jiang et al., 2021b). it contains about 12K images sampled from ImageNet-100 (Tian et al., 2020) with Pareto distribution. The instance number of each class ranges from 1,280 to 5. We use ImageNet-R (Hendrycks et al., 2021) as the OOD dataset. The dataset contains 30K images with several renditions (e.g., art, cartoons, deviantart) of ImageNet classes.

**Places-LT** The original Places (Zhou et al., 2017) is a large-scale scene-centric dataset. Places-LT (Liu et al., 2019) contains about 62.5K images sampled from Places with Pareto distribution. The instance number of each class ranges from 4,980 to 5. Places-Extra69 (Zhou et al., 2017) is utilized as the OOD dataset. It includes 98,721 images for 69 scene categories besides the 365 scene categories in Places.

**Training details** We implement all our techniques using PyTorch (Paszke et al., 2017) and conduct the experiments using RTX3090 GPUs. We evaluate our method with SimCLR (Chen et al., 2020) framework with batch size 512 for small datasets (CIFAR-10-LT/CIFAR-100-LT) and 256 for large datasets (ImageNet-100-LT/Places-LT) in default. We adopt Resnet-18 (He et al., 2016) for small datasets and Resnet-50 for large datasets, respectively. In our paper, we evaluate COLT's performance under two evaluation protocols in self-supervised learning: ***linear-probing*** and ***few-shot***. For both protocols, we first perform self-supervised training on the encoder model to get the optimized visual representation. Then, we fine-tune a linear classifier on top of the encoder (fixed during training the classifier). The only difference between linear probing and few-shot learning is we use the full dataset for linear probing and 1% samples of the full dataset for few-shot learning during fine-tuning. We keep all settings in the fine-tuning stage (e.g., optimizer, learning rate, batch size) the same as (Jiang et al., 2021b).

Mainly Following (Jiang et al., 2021b; Zhou et al., 2022; Jiang et al., 2021a), we pre-train all the baselines and COLT with 2000 epochs on CIFAR10/100, 1000 epochs on ImageNet-100, 500 epochs on Places. As for the fine-tuning stage, the "linear-probing" and "few-shot" results are produced by fine-tuning the classifier for 30 epochs and 100 epochs, respectively. To make a fair comparison, we implement COLT and all baselines with the same data augmentation strategies. We sample $K = 10,000$ OOD images on every $r = 25$ epoch for CIFAR-10-LT/CIFAR-100-LT, Places-LT, and $r = 50$ for ImageNet-100-LT.

## APPENDIX B  MORE EMPIRICAL RESULTS

We present the experiment results on ImageNet-100 with ImageNet-R or Places69 as the external OOD dataset in Table 6. We compare COLT with methods that make use of external data. MAK (Jiang et al., 2021a) is the state-of-the-art method that proposes a sampling strategy to re-balance the training set by sample in-distribution tail class instances. Note that "random" refers to randomly sampling from the external dataset according to the sampling budget. We observe both higher accuracy and balancedness under different sample budgets on different OOD datasets. It indicates COLT

---

[2] A debiased subset of 80 Million Tiny Images (Torralba et al., 2008) with 300K images. 80 Million Tiny Images is constructed by the corresponding images of 53,464 nouns from internet search engines.

leverages OOD data in a more efficient way. Besides, the performance gain of the minority classes (*Median* & *Few*) is more notable. Meanwhile, COLT yields balancedness feature space. Following previous works (Jiang et al., 2021b) (Zhou et al., 2022), we measure the balancedness of a feature space through the accuracy's standard deviation (Std) from *Many, Median* and *Few*. To further demonstrate that our proposed COLT is also effective on the non-curated open-world datasets, we conduct experiments on ImageNet-100-LT with a 50K subset of Open Images (Krasin et al., 2017) (a dataset of about 9 million images belonging to over 6000 categories.) as the OOD dataset. We can also observe a significant improvement in both the accuracy (especially for {Median, Few}) and the balancedness of the feature space.

Table 6: Accuracy (%) and balancedness (Std) on ImageNet-100 with different OOD datasets.

| Sampling pool | Budget | Method | Protocol | Many ↑ | Median ↑ | Few ↑ | Std ↓ | All ↑ |
|---|---|---|---|---|---|---|---|---|
| None | - | - | few-shot | 48.92 | 39.54 | 34.15 | 6.10 | 42.51 |
| | | | linear-probing | 69.53 | 63.74 | 59.88 | 3.97 | 65.50 |
| Places69 | 10K | random | few-shot | 52.97 | 44.03 | 39.98 | 5.43 | 46.99 |
| | | | linear-probing | 73.49 | 67.57 | 63.05 | 4.28 | 69.29 |
| | | MAK | few-shot | 54.16 | 45.17 | 41.43 | 5.34 | 48.19 |
| | | | linear-probing | 74.69 | 68.60 | 64.63 | 4.14 | 70.46 |
| | | COLT | few-shot | **55.82** | **48.32** | **43.49** | **4.83** | **49.24** |
| | | | linear-probing | **75.23** | **71.42** | **66.46** | **3.59** | **72.26** |
| Image Net-R | 5K | random | few-shot | 51.76 | 41.45 | 38.12 | 5.81 | 45.04 |
| | | | linear-probing | 72.22 | 66.49 | 63.45 | 3.64 | 68.33 |
| | | MAK | few-shot | 52.49 | 43.54 | 39.00 | 5.65 | 46.48 |
| | | | linear-probing | 73.24 | 67.62 | 64.14 | 3.75 | 69.36 |
| | | COLT | few-shot | **54.10** | **46.01** | **42.32** | **4.92** | **48.69** |
| | | | linear-probing | **74.52** | **69.67** | **67.53** | **2.92** | **71.28** |
| | 10K | random | few-shot | 52.82 | 42.88 | 40.27 | 5.41 | 46.42 |
| | | | linear-probing | 74.33 | 68.52 | 62.65 | 4.77 | 70.02 |
| | | MAK | few-shot | 54.33 | 45.01 | 40.12 | 5.90 | 48.01 |
| | | | linear-probing | **75.57** | 68.20 | 66.29 | 4.07 | 70.83 |
| | | COLT | few-shot | **54.26** | **46.54** | **43.38** | **4.57** | **49.14** |
| | | | linear-probing | 75.13 | **71.38** | 66.62 | 3.48 | 72.22 |

Table 7: Accuracy (%) and balancedness (Std) on ImageNet-100 with external 50K Open-Images (Krasin et al., 2017) as OOD dataset.

| Method | Budget | Protocol | Many ↑ | Median ↑ | Few ↑ | Std ↓ | All ↑ |
|---|---|---|---|---|---|---|---|
| SimCLR | - | few-shot | 49.74 | 41.00 | 36.62 | 5.45 | 43.84 |
| | | linear-probing | 70.82 | 65.33 | 62.31 | 3.52 | 67.08 |
| SimCLR+COLT | 5K | few-shot | 54.46 | 47.12 | 43.23 | 4.66 | 49.48 |
| | | linear-probing | 75.62 | 70.87 | 67.29 | 3.41 | 72.26 |
| | 10K | few-shot | 57.54 | 48.12 | 44.00 | 5.67 | 51.26 |
| | | linear-probing | 76.87 | 71.00 | 69.23 | 3.27 | 73.06 |

## APPENDIX C  HOW DO OOD DATA CONTRIBUTE TO LONG-TAIL LEARNING?

### OOD DATA RE-BALANCE THE FEATURE SPACE

As suggested in previous research, the standard contrastive learning would naturally put more weight on the loss of majority classes and less weight on that of minority classes, resulting in imbalanced

feature spaces and poor linear separability on tail samples(Kang et al., 2020; Li et al., 2022), i.e., the majority classes dominate the feature space and tail instances turn out to be outliers.

To quantitatively analyze the imbalance of the feature space, we define a metric called Normalized Misclassification Matrix (NMM):

$$\text{NMM}_{ij} = \frac{m_{ij}/\sum_{k=1}^{n} m_{ik}}{|\mathcal{T}_j|/\sum_{k=1}^{n}|\mathcal{T}_k|}, \tag{7}$$

where $\mathcal{T}_k$ is the $k$-th split of the long-tailed train set, satisfying $S_{id} = \cup_{k=1}^{n}\mathcal{T}_k$ and $\mathcal{T}_i \cap \mathcal{T}_j = \emptyset, \forall i \neq j$. In this paper, we follow the common practice in long-tail learning that split the dataset to $S_{id} = \mathcal{T} = \{\mathcal{T}_{\text{Few}}, \mathcal{T}_{\text{Median}}, \mathcal{T}_{\text{Many}}\}$ according to the instance number in each class, and $|\mathcal{T}_j|$ denote the class number in split $\mathcal{T}_j$. $m_{ij}$ represents the number of (misclassified) instances belonging to split $\mathcal{T}_i$ but are classified to split $\mathcal{T}_j$. Note that $m_{ii}$ indicates both the ground truth label and the (wrong) prediction fall into split $\mathcal{T}_i$. Intuitively, if the feature space is perfectly balanced (i.e., equal margin between different classes), each element in NMM is approximately 1.0, i.e., the misclassified samples nearly randomly fall into each split. On the contrary, higher (lower) mean margins between classes in $\mathcal{T}_i$ and $\mathcal{T}_j$ result in lower (higher) $m_{ij}$.

The results are shown in Fig 4. Fig 4a indicates when the train set is balanced, the mean margin of different classes is approximately equal (NMM$_{ij} \approx 1$), note that the split $\{\mathcal{T}_{\text{Few}}, \mathcal{T}_{\text{Median}}, \mathcal{T}_{\text{Many}}\}$ in Fig 4a is consistent with Fig 4b and Fig 4c. Fig 4b reflects that majority classes have a higher margin to other classes than minority classes. In other words, the model is more likely to confuse samples from a minority class with other minority classes, implying an imbalanced feature space. Fig 4c exhibits the result of COLT based on SimCLR with 10K OOD data. It's observed that COLT alleviates the imbalance issue since COLT augments minority classes with more instances which could be interpreted as an implicit loss re-weighting strategy.

OOD DATA BRIDGE INSTANCES FROM MINORITY CLASSES

In this section, we demonstrate the effect of OOD data on the perspective of contrastive learning. A recent work (Wang et al., 2021) gives a theoretical understanding of contrastive learning based on *augmentation overlap*. Concretely, they suggest that the samples of the same class could be very alike after aggressive data augmentations. Thus, the pretext task of aligning positive samples can facilitate the model to learn class-separable representations. They define the augmentation graph $\mathcal{G} = (\mathcal{V}, \mathcal{E})$ as: $N$ samples are the vertices of the graph, and there exists an edge $e_{ij}$ when sample $i$ and sample $j$ has overlapped views. According to their theory, intra-class augmentation overlap is a sufficient condition for gathering features from the same class. In this case, we compute the ratio of connected nodes (degree is not zero) to measure the extent of intra-class augmentation overlap in class $k$, denoted as score $s_c^k$. Ideally, each class should have $s_c^k = 1$, which indicates all samples from the same class will be clustered together during the contrastive training process. A smaller $s_c^k$ indicates lower intra-class consistency and vice versa.

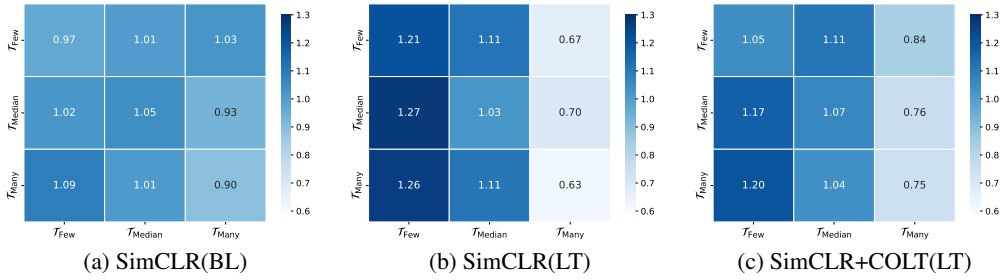

Figure 4: Normalized Misclassification Matrix (NMM) on the test set of CIFAR-100 with different frameworks and train sets. (4a): SimCLR trained on balanced CIFAR-100. (4b): SimCLR trained on long-tailed CIFAR-100. (4c): implement COLT on top of SimCLR trained on long-tailed CIFAR-100.

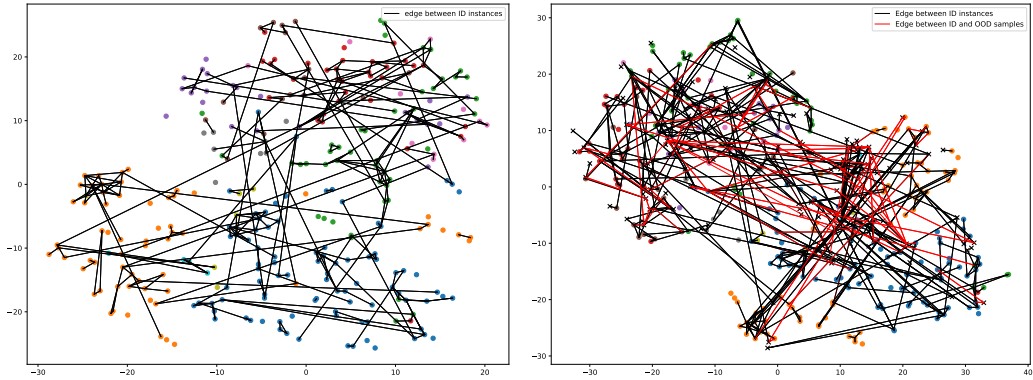

(a) SimCLR (majority classes' mean connectivity: 0.73 minority classes' mean connectivity: 0.46)

(b) SimCLR+COLT (majority classes' mean connectivity: 0.76 minority classes' mean connectivity: 0.67)

Figure 5: The augmentation graph of CIFAR-10. Similar to (Wang et al., 2021), We choose a random subset of test images and randomly augment them 20 times. Then, we calculate the instance distance in the representation space and draw edges for image pairs whose smallest view distance is below a small threshold. We visualize the samples with t-SNE and denote edges between ID instances in black and edges between ID and OOD samples, forming new connections in red.

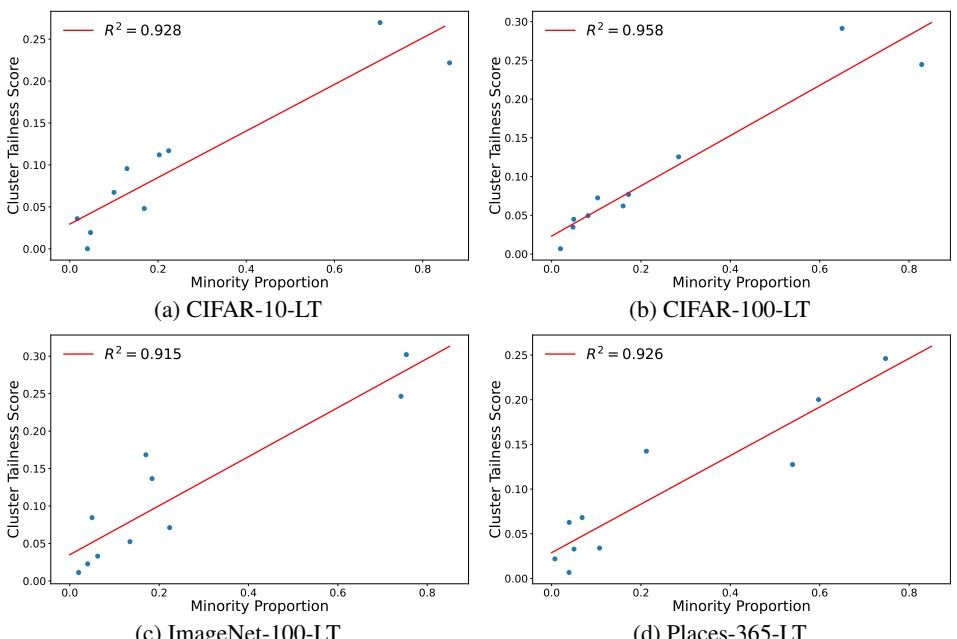

(a) CIFAR-10-LT

(b) CIFAR-100-LT

(c) ImageNet-100-LT

(d) Places-365-LT

Figure 6: Linear regression results between the minority proportion in a cluster and the cluster's tailness score on long-tailed CIFAR, ImageNet-100, and Places. We set cluster number $C = 10$.

We visualize the augmentation graph of CIFAR-10 in Fig 5 and compute the connectivity score for each class. We observe a smaller $s_c$ for minority classes than for majority classes, which is also consistent with the theoretical analysis in (Wang et al., 2021). Nevertheless, since we dynamically add OOD samples around ID (especially minority classes) samples, more instances from the ID minority class can be reachable on the augmentation graph via OOD samples (majority/minority connectivity improvement: 0.03/0.31). The red edges in Fig 5b illustrate that OOD samples help long-tail learning by bridging samples from minority classes, further improving intra-class consistency.

## APPENDIX D MORE ANALYSIS ABOUT COLT

**Different tail estimation strategies** In our paper, we defined a tailness score based on top-$k\%$ ($k = 2$ in practice) largest negative logits to localize the head and tail samples in an unsupervised manner. We also make a comparison to other alternative strategies. In Tab. 8, we provide results on locating tail samples by a radius-based definition (with radius as the sum of negative logits and select different radius thresholds) of tailness score or simply using the method in BCL (Zhou et al., 2022). The sampling budget is set to 5K, and the ID and OOD dataset is CIFAR-100-LT and 300K Random Images respectively. We can notice that there is no significant difference between COLT with Top-k% and radius-based methods, while both of them surpass BCL.

**Is the unsupervised clustering reliable?** In the paper, we propose to perform a K-means clustering to the samples from $S_{id}$, then calculate the cluster-wise sampling budget via the tailness score. Since our ultimate goal is to sample more (less) OOD data similar to the minority (majority) samples, it's natural to ask *how well does this clustering work*, *whether we assigned more budget to the tail samples*. To this end, we propose to measure the clustering and sampling quality by the ratio of minority samples in a cluster and its corresponding cluster-wise tailness score. Results are visualized in Fig 6. It's observed that the minority proportion of some clusters is close to 1, while others are composed of samples from majority classes. Besides, the cluster-wise tailness score shows a linear correlation of the minority proportion, implying we do allocate more sampling budget to tail classes according to Eq. 3.

**Impact of the cluster number** Recall in Sec 3.3 we first perform a K-means clustering to the ID samples, then select OOD samples close (with a large cosine similarity in the feature space) to the head or tail classes based on the budget allocation function. To verify 1), whether the cluster number will dramatically affect the performance 2), how the performance is when clustering based on the ground-truth label information rather than a self-supervised manner. We conduct experiments with various cluster number $C$ in Tab. 9, the OOD dataset is 300K Random Images (Hendrycks et al., 2018b), and the sampling budget is set to 5K. Note that the supervised clustering method is referred to as Oracle. The results show the performance of COLT will not be significantly affected by the number of clusters. Furthermore, cluster according to the label information (Oracle) achieves a similar performance compared to the unsupervised clustering, implying the K-means clustering can be used as a feasible alternative to roughly separates the minority from the majority classes in self-supervised scenarios.

**The scale / balancedness of the OOD dataset** In our work, we evaluate the proposed COLT on both balanced datasets, which are used as OOD datasets in previous works (Kumar et al., 2021; Wei et al., 2022) and non-curated open-world datasets such as 300K Random Images (Hendrycks et al., 2018b). Although COLT performs well on the aforementioned datasets, we intend to further ask *will COLT perform well when the OOD dataset is also long-tailed?* Tab. 10 shows the performance of COLT on CIFAR-100 when the OOD dataset (ImageNet-100) has various imbalance ratios. The similar accuracy suggests COLT is robust to the imbalancedness of the OOD dataset to some extent. This could be attributed to the dynamic sampling procedure filtering out most of the unhelpful OOD samples so that the performance will not be dramatically affected by changes in the external OOD dataset.

Another observation can be found from Tab. 11, where we measured how the OOD dataset's scale influences COLT. We form the external OOD dataset of different scales by gradually increasing the number of samples in Random 300K Images (Hendrycks et al., 2018b), and implementing COLT on those subsets of Random 300K Images. Conform to intuition, the scale of the OOD dataset is positively correlated with the performance of COLT since we may select more desired samples in a larger candidate set. However, the problem can be tricky when the dataset is also changed. For example, we observe a similar performance gain on ImageNet-100-LT when utilizing Places-69 (about 98K images) and ImageNet-R (30K images) as the auxiliary OOD dataset, which implies the scale of the OOD dataset is not the only factor affecting the performance.

Table 8: Comparison on different tail estimation strategies.

| Method | COLT | Radius-based | | | | | BCL |
|---|---|---|---|---|---|---|---|
| Threshold | - | 0.08 | 0.09 | 0.10 | 0.11 | 0.12 | - |
| Accuracy | $54.20 \pm 0.35$ | $53.19 \pm 0.47$ | $53.33 \pm 0.28$ | $54.00 \pm 0.21$ | $53.87 \pm 0.22$ | $53.0 \pm 0.34$ | $52.97 \pm 0.30$ |

Table 9: Ablation study on the cluster number $C$ on CIFAR-100-LT.

| Cluster Number C | 10 | 20 | 50 | 100 | 100 (Oracle) |
|---|---|---|---|---|---|
| Accuracy | $54.20 \pm 0.35$ | $53.61 \pm 0.18$ | $54.06 \pm 0.22$ | $53.81 \pm 0.29$ | $54.16 \pm 0.10$ |

Table 10: COLT's accuracy when OOD dataset is also imbalanced.

| Imbalance ratio | 1 | 10 | 50 | 100 |
|---|---|---|---|---|
| Accuracy | $52.98 \pm 0.33$ | $52.63 \pm 0.19$ | $52.74 \pm 0.25$ | $52.66 \pm 0.41$ |

Table 11: COLT's accuracy with different scale OOD dataset.

| OOD Images Number (K) | 100 | 200 | 300 |
|---|---|---|---|
| Accuracy | $53.43 \pm 0.26$ | $53.99 \pm 0.18$ | $54.20 \pm 0.35$ |

