# OpenReview forum: "On the Effectiveness of Out-of-Distribution Data in Self-Supervised Long-Tail Learning."
_ICLR.cc/2023/Conference — ICLR 2023 poster_

### Official Review · Reviewer_SHmZ · 2022-10-20

**Confidence:** 3
**Correctness:** 3
**Technical Novelty And Significance:** 2
**Empirical Novelty And Significance:** 3
**Recommendation:** 6

**Clarity, Quality, Novelty And Reproducibility:**

The authors design an effective method to help self-supervised long-tailed learning with the auxiliary of out-of-the-distribution data and demonstrate the promising performance on a range of datasets. My main concerns are about the technical novelty, which is relatively ad-hoc and some claims need to be refined.

**Strength And Weaknesses:**

Strength

(1) The experiments on a range of datasets demonstrate that it is promising to improve the performance of self-supervised long-tailed learning via external data.

(2) The proposed method is orthogonal to a range of existing methods and can be consistently improve the current self-supervised long-tailed learning methods.

There are some concerns that need to be improved.

(1) The first contribution in the Introduction is not very meaningful, since MAK has given such a setting and a rough answer to this problem. It is advised to refine this point, better claiming the challenges in this direction.

(2) The proposed method combines dynamic sampling guided by tailed area estimation and an additional supervised contrastive learning, which slightly makes the novelty incremental. Although it shows the additional supervised contrastive learning helpful in Figure 3(c), it also confuses the key challenge of this topic.

(3) The external datasets are also relatively controversial, since it seems that the authors also choose the datasets tightly in the same domain, e.g., ImageNet-100 with ImageNet-R, instead of the non-curated open-world datasets. Here, some details are not clear. What is 300K random Images in Hendrycks 2018b. for CIFAR-LT.

(4) It is not proper to roughly compare the computational overhead between COLT and MAK, since the periodic sampling on the basis of the clustering is also expensive. It might be more reasonable to compare the time cost relative to the plain backbone without the sampling augmentation.

**Summary Of The Paper:**

This paper studies self-supervised long-tailed learning with the auxiliary of out-of-distribution data by designing a sampling strategy together with a distribution-level supervised contrastive loss. Empirical validation demonstrates the effectiveness of the proposed method on a range of datasets compared with previous methods.

**Summary Of The Review:**

It is an interesting and promising work, and the technical novelty and some claims should be improved. I can consider to raise the score if above concerns can be well addressed.

---

> ### Author Response · Authors · 2022-11-15
> **Response to Reviewer SHmZ (2/2)**
>
> > The external datasets are also relatively controversial, since it seems that the authors also choose the datasets tightly in the same domain, e.g., ImageNet-100 with ImageNet-R, instead of the non-curated open-world datasets. Here, some details are not clear. What is 300K random Images in Hendrycks 2018b. for CIFAR-LT.
>
> Thanks for the valuable comment. In our experiments, we mainly follow existing papers for OOD dataset selection. In particular, we select ImageNet-R (composed of 30K images with several renditions, e.g., art, cartoons, deviantart, of ImageNet classes) as the OOD dataset for ImageNet-100 by following [1]. Following [2], we use 300K Random Images as the OOD dataset for CIFAR-10/CIFAR-100, Places-69 for Places-365.
>
> In our work, we also select less organized or non-curated datasets to simulate the scenarios in practice, such as Random 300K Images[3] for CIFAR-10/100. In particular, Random 300K Images[3] is a subset of 80 Million Tiny Images[4], and the latter is constructed by the corresponding images of 53,464 nouns from internet search engines. Hence, it can be considered as the **"non-curated open-world"** dataset, and we added a description of this dataset in the revision.
>
> Following your suggestion to further demonstrate that our proposed COLT is also effective on the non-curated open-world datasets, we conduct experiments on ImageNet-100-LT with a 50K subset of OpenImage[5] (a dataset of about 9 million images belongs to over 6000 categories.) as the OOD dataset. The results are as follows (also presented in Appendix B). We can observe a significant improvement in both the accuracy (especially for {Median, Few}) and the balancedness of the feature space (smaller std).
>
>
> | Method           | Budget  | Protocol | Many $\uparrow$ | Median $\uparrow$| Few $\uparrow$| Std $\downarrow$| All $\uparrow$|
> |:-----------------|:-------:|:-------:|:-----:|:------:|:--:|:-------:|:-------:|
> | SimCLR    | - |few-shot| 49.85 | 41.67  |37.08|5.28|44.26|
>  | SimCLR             | - |linear-probing| 69.53 | 63.74  |59.88|3.97|65.50|
> | SimCLR+COLT             | 5K |few-shot| 54.56 | 47.12  |43.23 |4.66|49.48|
> | SimCLR+COLT           | 5K |linear-probing| 75.62 | 70.87  |67.29|3.41|72.26|
> | SimCLR+COLT | 10K |few-shot| 57.54 | 48.12 |44.00|5.67|51.26|
> | SimCLR+COLT           | 10K |linear-probing| 76.87 | 71.00 |69.23|3.27| 73.06 |
>
> >It is not proper to roughly compare the computational overhead between COLT and MAK, since the periodic sampling on the basis of the clustering is also expensive. It might be more reasonable to compare the time cost relative to the plain backbone without the sampling augmentation.
>
> Thanks for the advice. We modified the discussion in the revision.
>
> The main advantage of COLT is that the time of performing sampling shall be less than training for one epoch in most cases. Therefore, in the text (before revised), we only roughly compare the computational overhead of COLT and MAK according to the number of training epochs. In fact, the computing cost of COLT is still significantly less than that of MAK, even if we count the sampling time. Take ImageNet-100-LT as an example: MAK first trains 700 epochs, then samples on the external dataset, and then retrains for 1000 epochs on the augmented dataset; while COLT trains 1000 epochs in total and re-samples every 100 epochs. The overhead of performing nine times sampling operations is much less than that of training on the ID images for 700 epochs.
>
> Note that we only do K-means clustering and calculate the similarity matrix during sampling. In each epoch, it's only needed to update the tailness score $s_{t}^{i}$ of each ID sample $x_i$ according to $ s_{t}^{i} = \sum_{top-k} p_{i}^{-} $, leading to small computational overhead.
>
> [1] Fine-tuning can distort pretrained features and underperform out-of-distribution.
>
> [2] Open-Sampling: Exploring Out-of-Distribution data for Re-balancing Long-tailed datasets.
>
> [3] Deep anomaly detection with outlier exposure.
>
> [4] 80 million tiny images: A large data set for nonparametric object and scene recognition.
>
> [5] OpenImages: A public dataset for large-scale multi-label and multi-class image classification. Dataset available from https://github.com/openimages

---

> ### Author Response · Authors · 2022-11-15
> **Response to Reviewer SHmZ (1/2)**
>
> We thank the reviewer for the insightful comments, and we address the questions below:
>
> > The first contribution in the Introduction is not very meaningful, since MAK has given such a setting and a rough answer to this problem. It is advised to refine this point, better claiming the challenges in this direction.
>
> We'd like to thank the reviewer for the constructive comments. We will revise the introduction accordingly.
>
> Both MAK and our COLT make use of external data to improve SSL long-tail learning. But we would like to highlight the main differences between our proposed COLT and the MAK:
>
> 1. MAK assumes external **ID samples are available**, while we consider a more practical scenario where **we only have access to OOD data** that can be easily collected (e.g., downloaded from the internet). In MAK, it rejects the OOD samples during training and views OOD samples as useless to performance improvement, while our COLT provides counter-intuitive insights to MAK, i.e., OOD samples could be beneficial to long-tail learning.
>
> 2. While ID samples in MAK could be naturally assumed helpful for the long-tail task, leveraging OOD to improve SSL long-tail learning is much more challenging since **OOD samples can be toxic** to the ID performance. For example, we observed a performance degradation when incorporating 5K images from 300K Random Images by random sampling into CIFAR-100-LT and implementing SimCLR on the augmented train set (from 47.65% to 45.94%). However, COLT can eliminate the toxicity of OOD samples and bring significant performance gain (from 47.65% to 54.20%).
>
> > The proposed method combines dynamic sampling guided by tailed area estimation and an additional supervised contrastive learning, which slightly makes the novelty incremental. Although it shows the additional supervised contrastive learning helpful in Figure 3(c), it also confuses the key challenge of this topic.
>
> Thanks for this insightful comment. We agree that the performance improvement is not very significant with the supervised contrastive learning loss. The reason for adding the SCL is as follows. Since the ID data usually exhibit a significant domain gap with OOD data, we propose the SCL loss to prevent the feature representation learned in the pre-training period from being utterly dominated by the external OOD samples. On the one hand, introducing a massive number of OOD samples close to tail classes can effectively balance the feature space. On the other hand, because the number of OOD data is much larger than the samples of ID data, it may harm the ID domain-specific features, which might further degrade the performance. Our proposed SCL can be treated as an ID/OOD classification loss to assist the network in distinguishing the ID/OOD samples. Moreover, we effectively balance the impact of OOD samples to get better performance by adjusting the coefficients of SCL and InfoNCE.

---

> > ### Author Response · Authors · 2022-11-19
> > **Further Response to Reviewer SHmZ**
> >
> > Following your suggestions, we refined the introduction to clarify the settings and challenges in *leveraging OOD data to help self-supervised long-tail learning*. We complemented the performance of the proposed COLT with OpenImage[1], a non-curated open-world dataset (Table 7 in Appendix B). Also, we refined the comparison of the computational overhead between COLT and MAK in Section 4.3. Please kindly refer to the revision.
> >
> > #### Technical novelty
> >
> > First, we raise a counter-intuitive but meaningful research direction on ***how to make the OOD data contribute the SSL learning in long-tailed tasks***. It origins from a non-trivial observation: directly incorporating OOD samples into self-supervised learning (Fig 3b) will lead to performance degradation (or slight improvement). To solve it, technically, we propose a novel SSL framework composed of tail estimation, dynamic sampling, and an auxiliary distribution-level loss to leverage OOD samples with consistent and significant performance improvements.
> >
> > To the best of our knowledge, our work presents the first attempt to extend additional training samples from OOD datasets for improved self-supervised long-tailed learning. The proposed SSL framework may have a boarder impact on the community.
> >
> > [1] OpenImages: A public dataset for large-scale multi-label and multi-class image classification. Dataset available from https://github.com/openimages

---

> ### Author Response · Authors · 2022-11-23
> **Thanks for Your Review**
>
> Dear Reviewer:
>
> Thanks again for the constructive comments and the time you dedicate to the paper! Many improvements have been made according to your suggestions and other reviewers' comments. We also updated the manuscript and summarized the changes above. We hope that our responses below and the revision will address your concerns. We would be grateful if you could take some time to review our responses and the revised manuscript. We are glad to follow up with your further comments.
>
> Thanks a lot again, and with sincerest best wishes
>
> Authors

---

### Official Review · Reviewer_QM2L · 2022-10-23

**Confidence:** 5
**Correctness:** 3
**Technical Novelty And Significance:** 2
**Empirical Novelty And Significance:** 3
**Recommendation:** 6

**Clarity, Quality, Novelty And Reproducibility:**

## Clarity
Fair.

x) The language is fine, but needs to be further improved.

x) The structure and flow are ok.

x) There are many typos in the text, I suggest a careful rereading.

Please refer to the weaknesses part for a complete review.


## Quality
Fair.

The paper is based on an idea that I find nice, but I feel that the paper could have developed and explored it more in depth.

## Novelty
Good.

The main message of the paper is a recipe to use OOD samples for improved long-tailed learning performance, and to exploit this knowledge for better training. As far as I know, this is novel.

## Reproducibility
Good.

The source code is provided with detailed running setups; although I still insist that more implementation details should be provided in the main paper, such as the training epochs, augmentation choices, etc.

Disclaimer: I did not check the code in detail.

**Strength And Weaknesses:**

# Strength
+ The investigation of OOD data for improving long-tailed recognition is under explored. The problem itself is important and practical, and the technique introduced is well motivated and clearly stated.
+ The experiments are thorough and cover large-scale benchmark datasets in the field. The performance is good -- compared to other CL baselines, the improvements on certain datasets are even larger than 10%.
+ The ablation studies are thorough. They confirm that each of the components should be useful for long-tailed data.
+ The analysis and visualization of the feature balanceness / ability for tail discovery are effective and clear.

---
# Weaknesses

Unfortunately, there are several major weaknesses that exist in the current paper.

### Methods & Experiments

- One drawback is that the paper fails to compare with semi-supervised learning methods for long-tailed recognition.
1. The main contribution of the paper is to consider OOD dataset in long-tailed learning, and design SSL algorithm to incorporate the OOD samples into the training process. However, one natural choice when given more unlabeled data is to use *semi-supervised* learning. For example, how does COLT compare to SOTA semi-supervised learning methods [1-3] when leveraging OOD data?

[1] FixMatch: Simplifying Semi-Supervised Learning with Consistency and Confidence.

[2] FlexMatch: Boosting Semi-Supervised Learning with Curriculum Pseudo Labeling.

[3] SelfMatch: Combining Contrastive Self-Supervision and Consistency for Semi-Supervised Learning

2. Moreover, in the long-tailed literature, methods have already been proposed to use unlabeled data for combating data imbalance [4-6]. Although they might use ID samples, it is still encourgaed to compare with these standard methods in the field to demonstrate the performance is convincing.

[4] CReST: A Class-Rebalancing Self-Training Framework for Imbalanced Semi-Supervised Learning. CVPR 2021.

[5] ABC: Auxiliary Balanced Classifier for Class-imbalanced Semi-supervised Learning

[6] Distribution aligning refinery of pseudo-label for imbalanced semi-supervised learning

- Another part I find confusing is the choice of OOD dataset.
1. How does OOD defined in the first place? Does it refer to covariate shift from the original data? Is the OOD strength controlled across different datasets? Overall, the authors did not give a principled way to define / contruct the OOD dataset.
2. Is the OOD dataset used also imbalanced / long-tailed? Will the imbalance of OOD dataset influence the results of COLT?  If so, how does it influence the performance? Unfortunately the analysis is missing here.
3. What is the amount (i.e., # of samples) of the OOD dataset? Will the number of OOD samples influence the performance?

- It is unclear how many epochs needed for training with COLT. In fact, no descriptions are provided for the actual training details. Does COLT need more training epochs, similar to contrastive learning methods like SimCLR / MoCo? What about training details for other baselines?

- Similar to above, contrastive learning-based methods usually use strong data augmentation tricks during training. However, many current methods do not use them, making the comparison unfair. Does COLT involve strong augmentations during training? This again goes to the comment that no detailed training setups are provided. If so, the authors should comprehensively compare COLT with current non-contrastive learning-based methods (such as RIDE) by adding such tricks to them.

- The proposed method can obtain enhancement on all many / medium / few-shot classes (e.g., on ImageNet-100-LT), which is perhaps surprising and a great benefit, but also leads to many questions. Many current methods sacrifice the head performance. Can the author explain where the benefit comes from, and why the current methods cannot achieve this goal?

### Writing
- The overall writing quality is not good. The writing needs to be significantly improved.
- The references in the text are somehow mixed with the main text without clear separation, making it really hard to read, especially for the related work. Please fix accordingly.
- There are many typos in the text, I suggest a careful rereading. Here are some examples:
  - page 7, "about 12% for longtail CIFAR-100" -> "about 12% for long-tailed CIFAR-100"
  - page 7, "COLT yields balancedness feature space" -> "COLT yields a balanced feature space"
  - page 8, "but also significantly alleviates the imbalancedness" -> "but also significantly alleviates the imbalance"

---
# Other questions
- The authors showed that using a distribution-level supervised contrastive loss to separate ID / OOD samples can lead to better performance. I do not fully understand the intuition here. The OOD samples used in the experiments are still largely overlapped with the ID samples (e.g., Places-LT uses still
"places" samples as OOD dataset; the contents do not change). That being said, the high-level semantic information are still there. Directly separating them may not lead to better feature space as many of the features can be shared across ID / OOD samples. Can you explain the rationale behind using this SCL loss?
- The distribution-level supervised contrastive loss is essentially try to discriminate between two sets of samples (ID or OOD). I wonder why not directly using a discriminator loss? What would be the differences?

**Summary Of The Paper:**

This paper studies self-supervised learning (SSL) for long-tailed datasets. In particular, it proposes to leverage out-of-distribution (OOD) data to improve model performance when facing imbalanced data, with a new framework proposed called Contrastive with Out-of-distribution (OOD) data for Long-Tail learning (COLT). It first proposes to localize tail samples by assigning a tailness score to each OOD sample, and then uses an online OOD sampling strategy to dynamically re-balance the feature space. Finally, a distribution-level supervised contrastive loss is proposed. The paper performs experiments on several long-tailed datasets and verifies that COLT improves the performance over other baselines.

**Summary Of The Review:**

The paper studies an interesting yet important problem, utilizing OOD samples for long-tailed learning. The paper proposed several interesting components to form a self-supervised learning framework called COLT, which is later demonstrated to deliver better results.

The overall idea of leveraging OOD data seems to be novel and not explored before in this field, which is plausible.

However, currently there are several drawbacks and weaknesses, in terms of Methods and Exepriments. The writing quality also needs to be significantly improved.

In summary, in my view, the current elaboration and presentation do not meet the standards of ICLR. I am open to increase my evaluation if my principal concerns in the Weaknesses section are addressed in a convincing manner.

---

> ### Author Response · Authors · 2022-11-15
> **Response to Reviewer QM2L (4/4)**
>
> ### About the supervised contrastive loss
>
> > The authors showed that using a distribution-level supervised contrastive loss to separate ID / OOD samples can lead to better performance. I do not fully understand the intuition here. The OOD samples used in the experiments are still largely overlapped with the ID samples (e.g., Places-LT uses still "places" samples as OOD dataset; the contents do not change). That being said, the high-level semantic information are still there. Directly separating them may not lead to better feature space as many of the features can be shared across ID / OOD samples. Can you explain the rationale behind using this SCL loss?
>
> Thanks for the insightful comment. Since the ID data usually exhibit a significant domain gap with OOD data, we propose the SCL loss to prevent the feature representation learned in the pre-training period from being utterly dominated by the external OOD samples. On the one hand, introducing a massive number of OOD samples close to tail classes can effectively balance the feature space. On the other hand, because the number of OOD data is much larger than the samples of ID data, it may harm the ID domain-specific features, which might further degrade the performance. Our proposed SCL can be treated as an ID/OOD classification loss to assist the network in distinguishing the ID/OOD samples. Moreover, we effectively balance the impact of OOD samples to get better performance by adjusting the coefficients of SCL and InfoNCE.
>
> > The distribution-level supervised contrastive loss is essentially try to discriminate between two sets of samples (ID or OOD). I wonder why not directly using a discriminator loss? What would be the differences?
>
> We empirically find that the performance of adding a discriminator loss is slightly worse than the supervised contrastive loss, and it brings a slight increase in the number of parameters (for adding an auxiliary classifier). Therefore we implement COLT with SCL rather than a discriminator loss.
>
> ### Other concerns
>
> > The references in the text are somehow mixed with the main text without clear separation, making it really hard to read, especially for the related work. Please fix accordingly.
>
> We are sorry for the inconvenience. We have fixed this problem in the revision by replacing "\cite" with "\citep".
>
> > There are many typos in the text, I suggest a careful rereading.
>
> We would like to thank the reviewer for a very detailed review. We will fix the typos.
>
> > The paper is based on an idea that I find nice, but I feel that the paper could have developed and explored it more in depth.
>
> Thanks for the advice. We further provide two additional analyses to explain the contribution of OOD samples in the proposed method. A brief description is as follows, and details are in the revised Appendix C.
>
> 1. From the perspective of feature space balancedness, we observe that the feature space of pre-training on long-tail datasets is highly imbalancedness, i.e., the majority classes have a larger margin to other classes than the minority classes. Nevertheless, the proposed COLT alleviates the imbalance of feature space to a great extent.
>
> 2. From the perspective of contrastive learning, OOD samples also help the minority classes by including more augmentation overlap, i.e., improving the "$\mathcal{T}$-connectivity" which is proposed in [1]. Since we dynamically add OOD samples around ID (especially minority classes) samples, more instances from the ID minority class can be reachable on the augmentation graph via OOD samples. As a result, OOD samples facilitate the model to learn class-separable ID representations, thus improving the performance.
>
> [1] Chaos is a Ladder: A New Theoretical Understanding of Contrastive Learning via Augmentation Overlap.

---

> > ### Comment · Reviewer_QM2L · 2022-11-18
> > **Thanks for the additional experiments**
> >
> > I appreciate the efforts the authors made to address my concerns.
> >
> > The newly added results look good to me. My following concerns are well addressed by the new experiments:
> > - COLT outperforms standard semi-supervised methods
> > - COLT outperfroms semi-supervised imbalanced learning methods (although I did not find related results in the updated paper; I still insist that a discussion would be beneficial)
> > - The additional studies on the property of OOD datasets are plausible.
> >
> > I like the note on the comparisons between self-supervised & semi-supervised methods when extra OOD data is available. I would suggest the authors to incorporate this into the corresponding sections to explain why semi-supervised methods might perform worse. A natural follow-up question would be: what if you train semi-supervised methods with more epochs (e.g., if self-supervised methods use 200 eps + 200 eps for two stage, you train semi-supervised with 400 eps)?
> >
> > For other comments, I understand that certain aspects are unavoidable - e.g., the construction of OOD datasets (and how "OOD" they are). They might not be immediately addressed, but are important parts to understand the role and benefits of OOD data, which is the core part of the paper.
> >
> > I also noticed from other reviews that there are some recent relevant publications on the same topic. It seems this paper did not provide comparisons to those also explore OOD data for long-tailed problems. Would love to hear comments from the authors and other reviewers also.
> >
> > To wrap up, given the additional (good) experimental results and clarifications on the training details, I'm willing to increase my score to 6 - leaning toward the accept side.

---

> > > ### Author Response · Authors · 2022-11-19
> > > **Further Response to Reviewer QM2L (2/2)**
> > >
> > > > I also noticed from other reviews that there are some recent relevant publications on the same topic. It seems this paper did not provide comparisons to those also explore OOD data for long-tailed problems. Would love to hear comments from the authors and other reviewers also.
> > >
> > > The most relevant works to ours are MAK[1] and Open-Sampling[2]. We will make a detailed discussion of the differences between them and provide a comprehensive comparison of these methods below.
> > >
> > > ### Differences between COLT and MAK[1]
> > >
> > > MAK assumes **external ID samples are available**, while we consider a more practical scenario where **we only have access to OOD data** that can be easily collected (e.g., downloaded from the internet). In MAK, it rejects the OOD samples during training and views OOD samples as useless to performance improvement, while our COLT provides counter-intuitive insights to MAK, i.e., OOD samples could be beneficial to long-tail learning.
> > >
> > > Our setting is more challenging since **OOD samples can be toxic** to the ID performance, while ID samples in MAK could be naturally assumed helpful for the long-tail task. For example, we observed a performance degradation when incorporating 5K images from 300K Random Images by random sampling into CIFAR-100-LT and implementing SimCLR on the augmented train set (from 47.65% to 45.94%). However, COLT can eliminate the toxicity of OOD samples and bring significant performance gain (from 47.65% to 54.20%).
> > >
> > > ### Differences between COLT and Open-Sampling[2]
> > >
> > > Open-sampling also utilized OOD data, but performed in a supervised manner while not directly applicable to the SSL frameworks. Concretely, they suggest OOD samples can be used to re-balance the class prior, which is not suitable in self-supervised learning since there is no "class" information in the pre-training stage.
> > >
> > > We are distinct from them in both motivation and technical contributions. We raise a meaningful research direction on ***how to make the OOD data contribute the SSL learning in long-tailed tasks.*** Technically, we propose a novel SSL framework composed of tail estimation, dynamic sampling, and an auxiliary distribution-level loss to leverage OOD samples with consistent and significant performance improvements. To the best of our knowledge, our work presents the first attempt to extend additional training samples from OOD datasets for improved self-supervised long-tailed learning.
> > >
> > > ### Comparison between COLT and MAK[1], Open-Sampling[2]
> > >
> > > As for MAK, we provide a detailed comparison of both accuracy (Table 3&4) and computation overhead (Section 4.3), and the proposed COLT outperforms MAK on top-1 accuracy with less computation overhead. As for Open-sampling, we also make a discussion on the computation overhead in Section 4.3, we do not compare COLT with Open-sampling in the paper since self-supervised methods (COLT) have stronger baseline than supervised methods (Open-sampling), e.g., about 38% and 47% on CIFAR-100-LT for supervised and self-supervised baseline respectively. However, COLT does achieve a higher performance gain compared to Open-Sampling, implying we leverage OOD data more efficiently. The comparison is as follows:
> > >
> > > | Method           | Accuracy |
> > > |:-----------------|:--------:|
> > > | Supervised baseline    | $37.59$  |
> > >  | +Open-Sampling(300K)              | $40.26(\bf{+2.67})$  |
> > > | Self-Supervised baseline             | $47.65$  |
> > > | +COLT(10K) | $57.46(\bf{+9.81})$  |
> > >
> > > [1] Improving Contrastive Learning on Imbalanced Seed Data via Open-World Sampling.
> > >
> > > [2] Open-Sampling: Exploring Out-of-Distribution data for Re-balancing Long-tailed datasets.

---

> > > > ### Comment · Reviewer_QM2L · 2022-12-10
> > > > **Response to Authors**
> > > >
> > > > Thanks for the explanation. I acknowledge the additional clarifications and experiments that the authors provided during the discussion period. These altogether have significantly improved the overall quality of the paper.
> > > >
> > > > Again, (almost all of) these clarifications and additional experiments are important and need to be incorporated into the revised version of the paper. A detailed comparison to existing works is crucial to correctly position the paper to the literature, and provide a comprehensive picture to the audience.

---

> > > > > ### Author Response · Authors · 2022-12-11
> > > > > **Thanks for the advice**
> > > > >
> > > > > Thanks for the advice! We will incorporate these classifications and experiments into the final version(some of them are already in the revision).

---

> > > ### Author Response · Authors · 2022-11-19
> > > **Further Response to Reviewer QM2L (1/2)**
> > >
> > > Thanks again for the elaborate review! We will address your concerns below:
> > >
> > > > I like the note on the comparisons between self-supervised & semi-supervised methods when extra OOD data is available. I would suggest the authors to incorporate this into the corresponding sections to explain why semi-supervised methods might perform worse. A natural follow-up question would be: what if you train semi-supervised methods with more epochs (e.g., if self-supervised methods use 200 eps + 200 eps for two stage, you train semi-supervised with 400 eps)?
> > >
> > > Thanks for the advice! We will incorporate the analysis between self-supervised and semi-supervised methods into Section 4 ASAP. As for the training epochs, we would like to clarify that we have trained FixMatch[1] and FlexMatch[2] with more epochs. We implement them with the default settings that train for $2^{20}$ iterations with batch size 64 for labeled images and 7x64 for unlabeled images. Hence, the total inference image number during training is $64\times8\times2^{20} \approx 5\times10^8$. While COLT train on the augmented dataset (about 20K images) with 2000 epochs and fine-tuned with 30 epochs, that is, about $4\times10^7$ images in total, which is much smaller than FixMatch. As for ABC[3] and DARP[4], we will train them with more epochs to ensure the total number of images is the same as COLT and provide the results in the revision.
> > >
> > > > For other comments, I understand that certain aspects are unavoidable - e.g., the construction of OOD datasets (and how "OOD" they are). They might not be immediately addressed, but are important parts to understand the role and benefits of OOD data, which is the core part of the paper.
> > >
> > > Thanks for the insightful comments! We agree that the measurement of "OOD" is critical for understanding the value of the external OOD dataset. Although we haven't given a metric for the measurement, we provide a feasible way to select helpful OOD samples to re-balance the feature space. Furthermore, we empirically verify the effectiveness of COLT on extensive OOD datasets, including balanced & commonly used datasets and non-curated open-world datasets. We also notice a few works [5,6] are devoted to giving metrics on dataset distance. Inspired by them, we intend to evaluate the relevance of ID and OOD, which is our ongoing work.
> > >
> > > [1] FixMatch: Simplifying Semi-Supervised Learning with Consistency and Confidence.
> > >
> > > [2] FlexMatch: Boosting Semi-Supervised Learning with Curriculum Pseudo Labeling.
> > >
> > > [3] ABC: Auxiliary Balanced Classifier for Class-imbalanced Semi-supervised Learning.
> > >
> > > [4] Distribution aligning refinery of pseudo-label for imbalanced semi-supervised learning.
> > >
> > > [5] Geometric dataset distances via optimal transport.
> > >
> > > [6] Robust Optimal Transport with Applications in Generative Modeling and Domain Adaptation.

---

> > > > ### Author Response · Authors · 2022-11-19
> > > > **Further Response to Reviewer QM2L**
> > > >
> > > > Following your suggestions, we have incorporated the comparisons between self-supervised and semi-supervised methods into Section 4.4. Please kindly refer to the revision.

---

> ### Author Response · Authors · 2022-11-15
> **Response to Reviewer QM2L (3/4)**
>
> ### About training details
>
> > It is unclear how many epochs needed for training with COLT. In fact, no descriptions are provided for the actual training details. Does COLT need more training epochs, similar to contrastive learning methods like SimCLR / MoCo? What about training details for other baselines?
>
> In the original version, we omit the report of some hyper-parameters if it's unchanged compared to the baselines, e.g., SDCLR, BCL, and MAK.  We have added a more detailed report in the revision. We would like to declare that we ensure that the training settings (e.g., training epochs, augmentation strategies, batch size, learning rate) are consistent with the baselines to make a **fair** comparison.
>
> For the training epochs, we pre-train all the baselines and COLT with 2000 epochs on CIFAR10/100 (consistent with [1,2]), 1000 epochs on ImageNet-100 (consistent with [3]), 500 epochs on Places ([1,2,3] doesn't report the results on Places, so we simply set the training epochs to a reasonable number). As for the fine-tuning stage, the "linear-probing" and "few-shot" results are produced by fine-tuning the classifier for 30 epochs and 100 epochs respectively (consistent with [1]).
>
> > Similar to above, contrastive learning-based methods usually use strong data augmentation tricks during training. However, many current methods do not use them, making the comparison unfair. Does COLT involve strong augmentations during training? This again goes to the comment that no detailed training setups are provided. If so, the authors should comprehensively compare COLT with current non-contrastive learning-based methods (such as RIDE) by adding such tricks to them.
>
> To make a fair comparison, we implement COLT and all baselines with the same data augmentation strategies. As you suggest, the strong data augmentation tricks involved in contrastive learning may lead to unfairness when compared to non-contrastive methods (such as supervised methods, like RIDE). Even if we "add strong augmentations" to methods such as RIDE,  the comparison might still be unfair since SSL methods use a two-stage training strategy and are always trained with more epochs, resulting in SSL methods outperforming others by a large margin, e.g., SDCLR[1] achieve 55.48% accuracy on CIFAR-100-LT, while 49.1% for RIDE with 4 experts). Hence, we follow the common practice in [2,3] by only making comparisons to other SSL methods to ensure fairness.
>
> > The proposed method can obtain enhancement on all many / medium / few-shot classes (e.g., on ImageNet-100-LT), which is perhaps surprising and a great benefit, but also leads to many questions. Many current methods sacrifice the head performance. Can the author explain where the benefit comes from, and why the current methods cannot achieve this goal?
>
> As pointed out by [4, 5], the feature space of contrastive learning is dominant by the minority classes due to the imbalancedness of the training set. The proposed COLT significantly improves the performance of minority classes ({Median, Few}) by re-balance the feature space with dynamic sampling OOD data. Meanwhile, the majority classes also benefit by having more diverse negative samples from the filtered OOD data, though with a smaller budget. In general, the external usage of either OOD (our method) or ID (MAK[3]) data might bring performance improvement for all many / median/ few classes, as compared to traditional re-weighting[6] or re-sampling[7] methods, which involve a trade-off between head and tail classes.
>
> [1] Self-Damaging Contrastive Learning.
>
> [2] Contrastive Learning with Boosted Memorization." International Conference on Machine Learning.
>
> [3] Improving Contrastive Learning on Imbalanced Seed Data via Open-World Sampling.
>
> [4] Exploring Balanced Feature Spaces for Representation Learning.
>
> [5] Targeted Supervised Contrastive Learning for Long-Tailed Recognition.
>
> [6] Class-Balanced Loss Based on Effective Number of Samples.
>
> [7] Relay backpropagation for effective learning of deep convolutional neural networks.

---

> ### Author Response · Authors · 2022-11-15
> **Response to Reviewer QM2L (2/4)**
>
> ### About OOD datasets
> > How does OOD defined in the first place? Does it refer to covariate shift from the original data? Is the OOD strength controlled across different datasets? Overall, the authors did not give a principled way to define / construct the OOD dataset.
>
> We use the common definition of OOD in the field of domain adaptation and domain generalization: assume the image $x_i$ and label $y_i$ in training set $S_{id}$ is sampled from the distribution $ Pr_{id}(X,Y) $, i.e., $(x_i,y_i) \sim Pr_{id}(X,Y), S_{id}=\{\cup_{i=1}^{n}(x_i, y_i)\}$. The out-of-distribution data $(x_i^{\prime},y_i^{\prime})$ are data which sampled from other distribution $Pr_{ood}(X,Y)$ where $Pr_{id}(X,Y) \neq Pr_{ood}(X,Y)$.
>
> In our work, we select the OOD datasets by following the same settings in previous works. For instance, following [1], we use 300K Random Images as the OOD dataset for CIFAR-10/CIFAR-100, Places-69 for Places-365; and following [2], we use ImageNet-R as OOD for ImageNet-100. We agree that we have not given a principled way to construct an OOD dataset, instead, we design a principled method to sample desired OOD images from various OOD datasets (toy dataset or open-world dataset as the response to the question below and Review SHmZ Q2) to further boost the performance.
>
> > Is the OOD dataset used also imbalanced / long-tailed? Will the imbalance of OOD dataset influence the results of COLT? If so, how does it influence the performance? Unfortunately the analysis is missing here.
>
> Since the OOD data can be less organized or varies a lot (e.g., collected from the internet) in practice, we do not make any further assumptions on the OOD dataset (e.g., balanced or imbalanced, the number of classes) in our work. To simulate the scenarios in practice, we evaluate the performance of COLT with both well-curated datasets and open-world datasets (kindly refer to Fig 3a), such as Random 300K Images[3], which is a subset of 80 Million Tiny Images[4]. The latter is constructed by the corresponding images of 53,464 nouns from internet search engines.
>
> Following your suggestion, we conduct experiments to examine the performance of COLT on CIFAR-100 when the OOD dataset (ImageNet-100) has various imbalance ratios:
>
> | Imbalance Ratio |       1        |       10       |       50       |      100       |
> |:----------------|:--------------:|:--------------:|:--------------:|:--------------:|
> | Accuracy        | $52.98\pm0.33$ | $52.63\pm0.19$ | $52.74\pm0.25$ | $52.66\pm0.41$ |
>
> The similar accuracy suggests COLT is robust to the imbalancedness of the OOD dataset to some extent. This could be attributed to the dynamic sampling procedure filtering out most of the unhelpful OOD samples so that the performance will not be dramatically affected by changes in the external OOD dataset.
>
> > What is the amount (i.e., \# of samples) of the OOD dataset? Will the number of OOD samples influence the performance?
>
> We have added a detailed description of the OOD datasets involved in the experiments. Kindly refer to Appendix A in the revision for the scale of each dataset. Intuitively, the scale of the OOD dataset is positively correlated with the performance of COLT since we may select more desired samples in a larger candidate set. To validate the hypothesis, we form the external OOD dataset of different scales by gradually increasing the number of samples in Random 300K Images[3], and implementing COLT on those subsets of Random 300K Images. The results are as follows:
>
> | OOD Images Number (K) |      100       |      200       |      300       |
> |:----------------------|:--------------:|:--------------:|:--------------:|
> | Accuracy              | $53.43\pm0.26$ | $53.99\pm0.18$ | $54.20\pm0.35$ |
>
> However, the problem can be tricky when the dataset is also changed. For example, we observe a similar performance gain on ImageNet-100-LT when utilizing Places-69 (about 98K images) and ImageNet-R (30K images) as the auxiliary OOD dataset, which implies the scale of the OOD dataset is not the only factor affecting the performance.
>
> [1] Open-Sampling: Exploring Out-of-Distribution data for Re-balancing Long-tailed datasets.
>
> [2] Fine-tuning can distort pretrained features and underperform out-of-distribution.
>
> [3] Deep anomaly detection with outlier exposure.
>
> [4] 80 million tiny images: A large data set for nonparametric object and scene recognition.

---

> ### Author Response · Authors · 2022-11-15
> **Response to Reviewer QM2L (1/4)**
>
> We thank the reviewer for the detailed feedback on our paper!
>
> > 1. The main contribution of the paper is to consider OOD dataset in long-tailed learning, and design SSL algorithm to incorporate the OOD samples into the training process. However, one natural choice when given more unlabeled data is to use semi-supervised learning. For example, how does COLT compare to SOTA semi-supervised learning methods [1-3] when leveraging OOD data?
>
> Thanks for this valuable comment. Following your suggestion, we implement FixMatch[1] and FlexMatch[2] on long-tailed CIFAR-100 with an imbalance ratio of 100, and compare the results in such semi-supervised learning scenarios (labeled: CIFAR-100-LT, unlabeled: 300K Random Images[3]) to supervised, self-supervised and our COLT methods. Note that we only replace the WRN-28-8 with ResNet-18 (also used in COLT) and keep other settings unchanged for a fair comparison. The results are as follows:
>
> | Method           | Accuracy |
> |:-----------------|:--------:|
> | ID-Supervised    | $44.13$  |
>  | FixMatch         | $47.38$  |
> | FlexMatch        | $50.40$  |
> | SimCLR           | $47.65$  |
> | SimCLR+COLT(10K) | $57.46$  |
>
> It can be observed that 1), external unlabeled OOD data can also be helpful when performing semi-supervised learning 2), the performance gains of COLT (about 10% on SimCLR) are more significant than incorporating OOD data via semi-supervised training (3.25% improvement on FixMatch and 6.27% on FlexMatch).
>
> In addition, the comparison between semi-supervised methods and COLT might be "unfair" to some extent. Although they both involve unlabeled OOD data, we assume the ID samples are label-agnostic during the pre-training stage while semi-supervised methods make use of the ID label information. However, self-supervised methods use a two-stage training strategy and are always trained with more epochs, resulting in SSL methods outperforming semi-supervised baselines by a large margin. Overall, our work mainly focuses on leveraging OOD data to improve the performance of **self-supervised** long-tailed learning, and we consider semi-supervised learning with OOD data as an open question for future exploration.
>
> > 2. Moreover, in the long-tailed literature, methods have already been proposed to use unlabeled data for combating data imbalance [4-6]. Although they might use ID samples, it is still encouraged to compare with these standard methods in the field to demonstrate the performance is convincing.
>
> Thanks for the suggestion. Following your suggestion, we implement ABC[4] and DRAP[5] on long-tailed CIFAR-100 with an imbalance ratio of 100. We set all ID samples (CIFAR-100-LT) labeled and OOD samples (300K Random Images[3]) unlabeled and replace the WRN-28-8 with ResNet-18 and keep other settings unchanged. We report the top-1 accuracy as follows:
>
> | Method           | Accuracy |
> |:-----------------|:--------:|
> | ID-Supervised    | $40.97$  |
>  | ABC              | $47.58$  |
> | DARP             | $46.60$  |
> | SimCLR           | $47.65$  |
> | SimCLR+COLT(10K) | $57.46$  |
>
> The proposed COLT achieves more significant performance improvement  (about 10%) compared to the semi-supervised methods designed for long-tail learning (about 7%), implying COLT leverages the external OOD data in a more efficient way.
>
> [1] FixMatch: Simplifying Semi-Supervised Learning with Consistency and Confidence.
>
> [2] FlexMatch: Boosting Semi-Supervised Learning with Curriculum Pseudo Labeling.
>
> [3] Deep anomaly detection with outlier exposure.
>
> [4] ABC: Auxiliary Balanced Classifier for Class-imbalanced Semi-supervised Learning.
>
> [5] Distribution aligning refinery of pseudo-label for imbalanced semi-supervised learning.

---

### Official Review · Reviewer_9FQm · 2022-10-24

**Confidence:** 3
**Correctness:** 3
**Technical Novelty And Significance:** 3
**Empirical Novelty And Significance:** 3
**Recommendation:** 8

**Clarity, Quality, Novelty And Reproducibility:**

The paper is clearly written, contains technical novelty, while its results seem reproducible. I encourage the authors to open source their code for better reproducibility.

**Strength And Weaknesses:**

Strengths:

- Self-supervised Learning (SSL) on long-tail data is a very important and interesting problem; training on balanced datasets like imagenet is highly artificial in that regard.

- Improving SSL on long-tailed datasets effectively with external unlabeled OOD data is a realistic scenario that was shown to work for supervised learning by Wei et al (2022).

- The method seems to give strong gains for SimCLR for long-tail SSL.

Weaknesses:

- The authors say that they "obtain C feature prototypes from ID training set S_id via K-means clustering.". How does one expect to capture a long-tail distribution with kmeans? How well does this clustering work, also when looking at clustering accuracy for the head and tail classes? Also, how do you set K? Is there a further assumption that we know how many classes exist?

- Defining the "tailness" score as simply the sum of the " top-k%"  largest negative logits seems like an unstable heuristic. Does k transfer across datasets? Did you consider other alternatives, eg radius-based?

- Although tackling a realistic problem in theory, the datasets used are mostly toy-level. Results on a real long-tail dataset like iNaturalist would be more convincing.

Notes:
- It is hard to undestand which line is which in Fig 3e, please fix the typo of two methods having the same line color and style

**Summary Of The Paper:**

The paper presents a method where out-of-domain (OOD) data are used to learning better representations via self-supervised learning (SSL) in a setting where we have unlabeled in-domain (ID) data that follow a long-tail distribution. The paper proposed to define an unsupervised "tailness" score, while an online sampling strategy is proposed for sampling the OOD data mostly closer to the tail classes. Experiments are conducted on 4 long-tail datasets

**Summary Of The Review:**

The paper deals with an interesting task and offers a solution that to my knowledge is novel, works better than recent works on the datasets it is tested on. Adding more realistic evaluations on bigger and real long-tail datasets, as well as expanding on aspects like the use of k-means clustering or different, more robust criteria for the tailness loss would make me more positive towards acceptance. Looking forward to the authors' responses.

---

> ### Author Response · Authors · 2022-11-15
> **Response to Reviewer 9FQm (1/2)**
>
> We thank the reviewer for providing encouraging comments on our paper. We provide clarifications to the concerns below:
>
> > 1. The authors say that they ``obtain C feature prototypes from ID training set Sid via K-means clustering". How does one expect to capture a long-tail distribution with kmeans? How well does this clustering work, also when looking at clustering accuracy for the head and tail classes?
>
> We agree with the reviewer that it is very difficult for K-means to capture the long-tail distribution precisely. But the main purpose of conducting K-means on ID data in our method is to roughly separate the head and tail data into different clusters, based on which the budget of external OOD samples are ***dynamically*** allocated. Empirically (Figure 4 in Appendix), we find that tail data tends to drop into several clusters. Hence, we can use those clusters to discover more OOD samples close to tail classes. Moreover, we found that clustering ID samples via unsupervised clustering (implemented in COLT) brings a similar performance of applying clustering based on the ground-truth label information (referred to as Oracle).
>
> | Cluster Number C | 10 | 20 | 50  | 100 |  100 (Oracle)   |
> |:----|:----:|:----:|:----:|:----:|:---------------:|
> | Accuracy         | $54.20\pm0.35$ | $53.61\pm0.18$ | $54.06\pm0.22$ | $53.81\pm0.29$ | $54.16\pm0.10$  |
>
> We further add a quantitative analysis of the quality of K-means clustering as in Table 1 (detailed in Appendix D). Since our ultimate goal is to sample more (less) OOD data similar to the minority (majority) samples, we propose to measure the clustering and sampling quality by the ratio of minority samples in a cluster and its corresponding cluster-wise tailness score. Results are shown in Fig 4. It's observed that the ratio of the minority in some clusters is close to 1.0, while others are composed of samples from majority classes. Besides, the cluster-wise tailness score shows a linear correlation w.r.t the ratio of minority, implying that we do allocate more sampling budget to tail classes according to Eq. 3.
>
> > Also, how do you set K? Is there a further assumption that we know how many classes exist?
>
> I think you might mean the number of clusters C. We keep $C=10$ on different datasets or SSL frameworks. Since K-means is used to separate the majority samples from the minority samples, the class number prior is not required in the proposed method. The experiments with different numbers of $C$ in CIFAR-100-LT, as shown in Table 1, also indicate our COLT is insensitive to $C$.
>
> > 2. Defining the "tailness" score as simply the sum of the " top-k%" largest negative logits seems like an unstable heuristic. Does k transfer across datasets? Did you consider other alternatives, eg radius-based?
>
> Top-k% largest negative logits in a single batch may lead to some randomness to the tail estimation. Nevertheless, the momentum update strategy makes it much more reliable and stable after a few epochs and discovers more samples from the tail compared to BCL [1] (please refer to the revised version of Fig. 3e).
>
> > Does k transfer across datasets? Did you consider other alternatives, eg radius-based?
>
> We keep $k=2$ across different datasets. Thanks for the suggestion.  We tried to locate tail samples by a radius-based definition (with radius as the sum of negative logits and select different radius thresholds) of tailness score or simply used the method in BCL, the results are as follows (the sampling budget is set to 5K, the ID and OOD dataset is CIFAR-100-LT and 300K Random Images respectively). We can notice that there is no significant difference between COLT with Top-k% and radius-based methods, while both of them surpass BCL.
>
> Radius-based tail discovering with different thresholds:
>
> | Threshold |      0.08      |      0.09      |      0.10      |      0.11      |      0.12      |
> |:----------|:--------------:|:--------------:|:--------------:|:--------------:|:--------------:|
> | Accuracy  | $53.19\pm0.47$ | $53.33\pm0.28$ | $54.00\pm0.21$ | $53.87\pm0.22$ | $53.00\pm0.34$ |
>
> Comparison between different tail mining strategies:
>
> | Method   |      COLT      | Radius-based (threshold=0.10) |      BCL       |
> |:---------|:--------------:|:-----------------------------:|:--------------:|
> | Accuracy | $54.20\pm0.35$ |        $54.00\pm0.21$         | $52.97\pm0.30$ |
>
> [1] Zhou, Zhihan, et al. "Contrastive Learning with Boosted Memorization." International Conference on Machine Learning. PMLR, 2022.

---

> > ### Author Response · Authors · 2022-11-15
> > **Response to Reviewer 9FQm (2/2)**
> >
> > > 3. Although tackling a realistic problem in theory, the datasets used are mostly toy-level. Results on a real long-tail dataset like iNaturalist would be more convincing.
> >
> > Thanks for the advice. We have conducted experiments on iNaturalist, see below the results:
> >
> >
> > | Method      |  Many $\uparrow$   | Median $\uparrow$  |   Few $\uparrow$   |  Std $\downarrow$  |   All $\uparrow$   |
> > |:------------|:-------:|:-------:|:-------:|:------:|:-------:|
> > | SimCLR      | $60.73$ | $52.47$ | $48.28$ | $5.17$ | $51.67$ |
> > | SimCLR+COLT | $62.35$ | $55.71$ | $52.60$ | $4.07$ | $55.29$ |
> >
> > We per-train the ResNet-50 for 200 epochs and fine-tuned it for 90 epochs. We select ImageNet as the OOD dataset, and the sampling budget is set to 100K. We can see COLT  brings significant performance gains and alleviates the imbalances of feature space. We will include the results on iNaturalist more comprehensively in the future version when more time is allowed for experiments.
> >
> > > It is hard to undestand which line is which in Fig 3e, please fix the typo of two methods having the same line color and style.
> >
> > Thanks for pointing out the mistakes. We have corrected them in the revision.
> >
> > > I encourage the authors to open source their code for better reproducibility.
> >
> > We have uploaded our code anonymously. Please refer to the reproducibility statement before the appendix for the link.

---

> > > ### Comment · Reviewer_9FQm · 2022-11-19
> > > **Thank you for the response**
> > >
> > > I want to thank the reviewers for clarifying a number of my concerns. It is nice to see that results hold on iNaturalist, a far more realistic LT setting. I also enjoyed the added experiments and discussion below with  QM2L. Despite some issues that can be traced back to how does one effectively perform clustering on long-tail data, this paper offers a simple and seemingly effective strategy for an interesting and in my opinion important problem. I raise my score to accept.

---

> > > > ### Author Response · Authors · 2022-11-19
> > > > **Further Response to Reviewer 9FQm**
> > > >
> > > > Many thanks for your valuable and cheering feedback! We will definitely keep moving forward to address the limitations and provide a better theoretical understanding of how to leverage OOD for SSL long-tailed learning in the future.

---

### Official Review · Reviewer_EuL2 · 2022-10-25

**Confidence:** 5
**Correctness:** 3
**Technical Novelty And Significance:** 2
**Empirical Novelty And Significance:** 2
**Recommendation:** 6

**Clarity, Quality, Novelty And Reproducibility:**

- The citation format should be improved. In the current version, the citation is mixed with the text part, which makes it difficult to read. I suggest the authors to use \citep instead.

- The writing should be improved. As shown in the weakness, the authors should not simply describe the method without any in-depth understanding.

- Originality: From my perspective, the proposed method is novel but not interesting enough.

**Strength And Weaknesses:**

Strength:
- The idea in this work is novel. Before this work, It has not been shown how to exploit the benefits of OOD data for SSL long-tailed setting.
- The improvements shown in the reported results are significant. From the reported results, the proposed method can effectively utilize the benefits of OOD examples in the SSL long-tailed settings.


Weakness:
- The contribution of this work is limited. This work can be seen as an SSL version of the recent work [1], which has shown that OOD can be beneficial to long-tailed learning. Although this work provides some technical contributions by adapting OOD data for SSL methods, this work cannot provide new insight for this problem.
- This work does not provide any in-depth understanding about why OOD examples can improve SSL in long-tailed setting. In this paper, the authors simply propose a method with three components, but do not explain why we could use OOD examples there.


[1] Wei, H., Tao, L., Xie, R., Feng, L., & An, B. (2022, June). Open-Sampling: Exploring Out-of-Distribution data for Re-balancing Long-tailed datasets. In International Conference on Machine Learning (pp. 23615-23630). PMLR.




**Summary Of The Paper:**

In this paper, the authors aim to employ OOD data to improve self-supervised learning in the long-tailed setting.  To achieve that, they use tailness score estimation, dynamic sampling strategies, and additional contrastive losses for long-tailed learning with additional OOD samples.  The authors conduct experiments on various datasets and several state-of-the-art SSL frameworks to evaluate the effectiveness of the proposed method.

**Summary Of The Review:**

The proposed method is novel and has shown significant improvement from the reported results. However, this work does not provide any insight or understanding about why it can work. Since this work can be treated as an SSL version of the recent work [1], the contribution of this work is not sufficient for this conference. So, I recommend a weak reject for this work.

-----

The response from the authors has addressed my concerns so I decide to improve my score to 6.

---

> ### Author Response · Authors · 2022-11-15
> **Response to Reviewer EuL2**
>
> Thanks for your constructive comments on our work!
>
> > 1. The contribution of this work is limited. This work can be seen as an SSL version of the recent work [1], which has shown that OOD can be beneficial to long-tailed learning. Although this work provides some technical contributions by adapting OOD data for SSL methods, this work cannot provide new insight for this problem.
>
> Thanks for the comments. We agree that both our work and [1] leverage OOD data for the long-tail problem. However, we would like to claim that our work provides new insights and technical contributions compared to [1].
>
> First, we raise a counter-intuitive but meaningful research direction on ***how to make the OOD data contribute the SSL learning in long-tailed tasks.*** It origins from a non-trivial observation: directly incorporating OOD samples into self-supervised learning (Fig 3b) will lead to performance degradation (or slight improvement). To solve it, technically, we propose a novel SSL framework composed of tail estimation, dynamic sampling, and an auxiliary distribution-level loss to leverage OOD samples with consistent and significant performance improvements. In contrast, [1] suggests OOD samples can be used to re-balance the class prior, which is not suitable in self-supervised learning since there is no "class" information in the pre-training stage.
>
> To the best of our knowledge, our work presents ***the first attempt to extend additional training samples from OOD datasets for improved self-supervised long-tailed learning***. The proposed SSL framework may have a boarder impact on the community.
>
> Moreover, we provide non-trivial understandings of ***why OOD data can contribute to the learning process*** via the lens of feature space balancedness and ``$\mathcal{T}$-connectivity''[2], as the answer to the question below.
>
> In addition, [1] performed in a supervised manner while **not** directly applicable to the SSL frameworks. Concretely, they suggest OOD samples can be used to re-balance the class prior, which is not suitable in self-supervised learning since there is no "class" information in the pre-training stage.
>
> > 2. This work does not provide any in-depth understanding about why OOD examples can improve SSL in long-tailed setting. In this paper, the authors simply propose a method with three components, but do not explain why we could use OOD examples there.
>
> Thanks for the constructive comments. Our work has yielded sound results in experiments. Moreover, we provide two additional analyses to explain the contribution of OOD samples in the proposed method:
>
> 1. From the perspective of feature space balancedness, we observe that the feature space with pre-training on long-tail datasets is highly imbalanced, i.e., the majority classes have a larger margin to other classes than the minority classes. Nevertheless, the proposed COLT alleviates the imbalance of feature space to a great extent.
>
> 2. From the perspective of contrastive learning, OOD samples also help the minority classes by including more augmentation overlap, i.e., improving the ``$\mathcal{T}$-connectivity'' which is proposed in [2]. Since we dynamically add OOD samples around ID (especially minority classes) samples, more instances from the ID minority class can be reachable on the augmentation graph via OOD samples. As a result, OOD samples facilitate the model to learn class-separable ID representations, thus improving the performance.
>
> Please kindly refer to **Appendix C** for details.
>
> [1] Wei, Hongxin, et al. "Open-Sampling: Exploring Out-of-Distribution data for Re-balancing Long-tailed datasets." International Conference on Machine Learning. PMLR, 2022.
>
> [2] Wang, Yifei, et al. "Chaos is a Ladder: A New Theoretical Understanding of Contrastive Learning via Augmentation Overlap." International Conference on Learning Representations. 2021.

---

> > ### Comment · Reviewer_EuL2 · 2022-11-24
> > **Thank you for the response**
> >
> > Thank you for the detailed response, and it has basically addressed my concerns. So, I decide to improve my score to 6.

---

> > > ### Author Response · Authors · 2022-11-25
> > > **Further Response to Reviewer EuL2**
> > >
> > > Thanks for the reply and appreciation! We will continue to improve our work.

---

> ### Author Response · Authors · 2022-11-23
> **Thanks for Your Review**
>
> Dear Reviewer:
>
> Thanks again for the constructive comments and the time you dedicate to the paper! Many improvements have been made according to your suggestions and other reviewers' comments. We also updated the manuscript and summarized the changes above. We hope that our responses below and the revision will address your concerns. We would be grateful if you could take some time to review our responses and the revised manuscript. We are glad to follow up with your further comments.
>
> Thanks a lot again, and with sincerest best wishes
>
> Authors

---

### Author Response · Authors · 2022-11-16
**Summary of Revision**

We thank all the reviewers for their time and thoughtful feedback. We revised our manuscript and submitted a new version for review.

## Summary of Revision.

### Main Paper

1. We refine the introduction for a precise statement of the challenges. (Reviewer SHmZ)

2. We modify the discussion between the proposed COLT and MAK[1]. (Reviewer SHmZ)

3. We add a comparison to Semi-Supervised Methods in Section 4.4. (Reviewer QM2L)

4. We modify the format of citations to make it easier to read. (Reviewer EuL2)

5. We fix the line color and style in Fig 3e. (Reviewer 9FQm and QM2L)

6. We fix some typos.

### Appendix

1. Adding Training details (Appendix A): As suggested by reviewer QM2L, we give detailed training settings in addition to Section 4.1.

2. More empirical results (Appendix B): We provide results on ImageNet-100-LT with OpenImage[2] (a non-curated open-world dataset) as the external dataset. (Reviewers SHmZ)

3. Analysis of the role of OOD data (Appendix C): We make an in-depth analysis of the effectiveness of OOD data through the lens of feature space balancedness and ``$\mathcal{T}$-connectivity''[3]. (Reviewer EuL2 and QM2L)

4. More analysis of the proposed COLT (Appendix D):

$\qquad$ (1) Comparison between different tail estimation strategies. (Reviewer 9FQm)

$\qquad$ (2) Reliability of the unsupervised clustering. (Reviewer 9FQm)

$\qquad$ (3) Ablation on the cluster number $C$. (Reviewer 9FQm)

$\qquad$ (4) COLT performance on imbalanced or different scale OOD datasets. (Reviewer QM2L)

[1] Improving Contrastive Learning on Imbalanced Seed Data via Open-World Sampling.

[2] OpenImages: A public dataset for large-scale multi-label and multi-class image classification. Dataset available from https://github.com/openimages

[3] Chaos is a Ladder: A New Theoretical Understanding of Contrastive Learning via Augmentation Overlap.

---

### Author Response · Authors · 2022-11-18
**General Response to All Reviewers**

Dear Reviewers:

We would like to thank you for your time and insightful comments! We have comprehensively revised our paper according to your comments (please kindly refer to the revision summary below). We hope we have addressed your concerns regarding the motivation for the SCL loss, sampling strategy, training settings, etc. Since the discussion is about to close, we would be grateful if you would kindly let us know of any other concerns and if we could further assist in clarifying any other issues.

Thanks a lot again, and with sincerest best wishes

Authors

---

### Decision · Program_Chairs · 2023-01-20

**Decision:**

Accept: poster

**Justification For Why Not Higher Score:**

The paper is solid. However, the scope of the contribution is somewhat narrow, focusing on the problem of fixing the bias in representations learned with LT data.


**Justification For Why Not Lower Score:**

Intuitive idea, strong empirical results, well written paper.

**Metareview: Summary, Strengths And Weaknesses:**

The paper shows how to use out-of-domain (OOD) data to improve learning in the set up of long-tail recognition,
The argument is that the representations learned with LT data are biased against tail classes,
but OOD can be used to balance the feature distribution of the representation. The paper defines and assigns a "tail-ness" score to OOD samples based on their neighbors in the latent space. It then uses that score for sampling better the tail-like samples. The idea is intuitive, and empirical results are strong.

After a detailed rebuttal, where the authors provided more results and clarifications, all reviewers recommended accepting the paper.

**Note From Pc:**

if the above contains the word "oral" or "spotlight" please see: "oral" presentation means -> notable-top-5% and "spotlight" means -> notable-top-25%. As stated in our emails, we are disassociating presentation type from AC recommendations

**Summary Of Ac-Reviewer Meeting:**

N/A